# Bedymo: a combined quasi-geostrophic and primitive equation model in sigma coordinates

Clemens Spensberger, Trond Thorsteinsson, and Thomas Spengler

Geophysical Institute and Bjerknes Centre for Climate Research, University of Bergen, Bergen, Norway

**Correspondence:** Clemens Spensberger (clemens.spensberger@uib.no)

**Abstract.** This paper introduces the idealised atmospheric circulation model Bedymo, which combines the quasi-geostrophic approximation and the hydrostatic primitive equations in one modelling framework. The model is designed such that the two systems of equations are solved as similarly as possible, such that differences can be unambiguously attributed to the different approximations, rather than the model formulation or the numerics. As a consequence, but in contrast to most other quasi-geostrophic models, Bedymo is using sigma-coordinates in the vertical. In addition to the atmospheric core, Bedymo also includes a slab ocean model with options for prescribed and wind-induced currents. Further, Bedymo has a graphical user interface, making it particularly useful for teaching.

Bedymo is evaluated for four atmosphere-only test cases and one coupled test case including the slab ocean component. The atmosphere-only test cases comprise the growth of a cyclonic disturbance in a baroclinic environment and the excitation of Rossby waves and inertia-gravity waves by isolated orography, all in a mid-latitude channel, as well as the simulation of a mid-latitude storm track. The atmosphere-ocean coupled test case is based on an equatorial channel and evaluates the coupled response to an isolated equatorial temperature anomaly in the ocean mixed layer. For all test cases, results agree well with expectations from theory and results obtained with more complex models.

## 1 Introduction

Since the 1950s, several studies introduced different quasi-geostrophic (QG; e.g. Charney and Phillips, 1953) and hydrostatic primitive-equation models (PE; e.g. Smagorinsky, 1958). Table 1 of Schär and Wernli (1993) lists a more recent selection of QG and PE models, and the development of idealised models is still ongoing (e.g. Hogg et al., 2003; Maze et al., 2006). Given this wealth of existing models, why should one develop another model? The main reason for developing the BErgen DYnamical MOdel (Bedymo) is to combine two approximations – quasi-geostrophy and the dry hydrostatic primitive equations – in one modelling framework. To our knowledge, there is no other model that combines several approximations in one numerical framework. Hence, Bedymo provides the only dynamical core that incorporates two levels in the Held (2005) hierarchy of models.

Typically, the approach to solve the underlying set of equations is different for each approximation. For example, QG models usually forecast the QG potential vorticity (QGPV) and diagnose the geostrophic wind and temperature field by inverting the QGPV (e.g. Charney and Phillips, 1953). In contrast, in PE models the horizontal winds are forecasted independently rather

than diagnostically derived from geostrophy. This remains true even if many PE models forecast vorticity and divergence instead of the wind components directly (e.g. Smagorinsky, 1958). As our main goal is to combine the different approximations in one model, we devise a common approach based on prognostic equations for temperature and surface pressure. This approach allows to solve the respective set of equations as similarly as possible. In contrast to previous studies (e.g. Whitaker, 1993; Rotunno et al., 2000), we can then intercompare QG with PE without using different models.

The simplifications from QG to PE have far-reaching consequences for the representation of the mid-latitude atmospheric dynamics. While in general the synoptic-scale dynamics are relatively well represented in QG (Charney and Phillips, 1953), there are several aspects of synoptic-scale weather systems that QG cannot represent. First and foremost, QG cyclones cannot develop fronts (cf., Hoskins, 1975), which are otherwise the most dynamically active parts of a cyclone. Further, the neglect of advection by the ageostrophic winds makes QG cyclones and anticyclones much more symmetric than their real counterparts (Wolf and Wirth, 2015). Both fronts and cyclone–anticyclone asymmetries are well represented in a PE model.

From a more conceptual perspective, the difference between QG and PE is the difference between a model that solely represents a balanced component of the flow (QG), and a model that can support flow imbalances (PE). In QG, the model state is given by the distribution of a single variable (i.e., QGPV), all other variables are derived diagnostically from balance relations. In contrast, in PE the horizontal winds and temperature can evolve principally independently from each other. As a consequence, only a PE model can represent the unbalanced flow associated with inertia-gravity waves. The ability to switch between the QG and PE representations makes Bedymo thus ideally suited to address long-standing questions on the relation between the balanced and unbalanced flow components and the role of the unbalanced flow to keep the flow nearly balanced (Plougonven and Snyder, 2007; McIntyre, 2009; Plougonven and Zhang, 2014).

Another reason for starting the model development from scratch was to make use of comparatively new features of Fortran 95 and 2003. These features aid the modularity and readability, and hence maintainability of the source code. In addition to the combined QG/PE atmospheric model, Bedymo also includes a slab ocean and atmospheric tracers as optional modules. Furthermore, the source code is organised in a way to make it easily accessible from Python. Besides allowing for flexible, yet easy-to-read runscripts, the Python bindings provide the basis for a graphical user interface (GUI) that allows to interactively run the model and watch the flow evolution "live" while the model is running. A user guide is available (Spensberger, 2022b). All these features make Bedymo an ideal tool not only for research, but also for education and student research projects.

## 2 The model

### 2.1 Atmospheric dynamics

Our joint approach to solve the equations for QG and for hydrostatic PE is based on the thermodynamic equation, because it features only small modifications between the different approximations. The only difference is the wind velocity components used in the advection (Tab. 1). In conjunction with a lower boundary condition provided by the forecasted surface pressure, the temperature distribution fully determines the atmospheric state in the QG system via hydrostatic and geostrophic balance. In PE, the horizontal wind velocity components evolve independently and hence must be forecasted separately.

In both systems, pressure and $\sigma$ vertical velocity is required to integrate the thermodynamic equation forward in time. In QG, the pressure vertical velocity follows from an inversion of the omega-equation, which implicitly establishes the three-dimensional QG-balance, and $\sigma$-vertical velocity is derived therefrom. In PE, both vertical velocities are derived from the continuity equation, using the divergence of the forecasted horizontal flow and the local surface pressure tendency. In PE, the local surface pressure is given by the column-integrated mass flux divergence and thus also follows from continuity. In contrast, in QG the local surface pressure tendency is derived from the pressure vertical velocity at the lower surface, and thus by the lower boundary condition used for the $\omega$-inversion. This boundary condition is given by the vorticity equation evaluated at the lower surface such that in QG surface pressure evolves following QG vorticity dynamics.

In the following, we summarise the model equations for QG and PE. We use a $\beta$-plane approximation for the Coriolis parameter $f = f_0 + \beta y$. In order to derive a self-consistent QG system, we need to approximate the specific volumes by time-invariant and horizontally homogeneous background values $\overline{\alpha} = \overline{\alpha}(\sigma)$. This approximation is optional for PE, and we will introduce and evaluate both variants. All symbols used in the following equations are summarised in Tab. 2.

### 2.1.1 Full primitive equations

The PE equations, as used in Bedymo, are

$$\frac{du}{dt} - (f_0 + \beta y)v = -\frac{\partial \phi}{\partial x} - \alpha \sigma \frac{\partial p_s}{\partial x} - ru + D\nabla^2 u \tag{1}$$

$$\frac{dv}{dt} + (f_0 + \beta y)u = -\frac{\partial \phi}{\partial y} - \alpha \sigma \frac{\partial p_s}{\partial y} - rv + D\nabla^2 v \tag{2}$$

$$\alpha = \frac{RT}{\sigma p_s} \tag{3}$$

$$\frac{\partial \phi}{\partial \sigma} = -\frac{RT}{\sigma} \tag{4}$$

$$\frac{dT}{dt} - \frac{\alpha}{c_p}\omega = \frac{J}{c_p} \tag{5}$$

$$\frac{1}{p_s}\frac{dp_s}{dt} + \nabla_\sigma \cdot \boldsymbol{v} + \frac{\partial \dot{\sigma}}{\partial \sigma} = 0 \tag{6}$$

In order to provide an energy sink for long and short waves, respectively, the momentum equations include scale-inselective linear Ekman friction and a scale-selective damping term.

In this system, we infer geopotential $\phi$ by integrating the hydrostatic equation (4) upwards, starting from the time-invariant surface geopotential $\phi_s$ which represents the model orography. The local surface pressure tendency results from continuity,

$$\frac{\partial p_s}{\partial t} = -\int_0^1 \nabla \cdot (\boldsymbol{v}p_s) \, d\sigma \ . \tag{7}$$

And, finally,

$$\dot{\sigma} = -\frac{\sigma}{p_s}\frac{\partial p_s}{\partial t} - \frac{1}{p_s}\int_0^\sigma \nabla \cdot (\boldsymbol{v}p_s)\, d\sigma \text{ , and} \tag{8}$$

$$\omega = -\int_0^\sigma \nabla \cdot (\boldsymbol{v}p_s)\, d\sigma + \sigma\boldsymbol{v} \cdot \nabla p_s \text{ ,} \tag{9}$$

which closes the system of equations (1)-(6). All other parameters in the equations are either time-invariant fields, constants, or represent an external forcing.

### 2.1.2   Homogeneous density approximation

As a first step towards quasi-geostrophy, we first assume horizontally homogeneous density within the PE system. In mathematical form, the approximation $\alpha \approx \overline{\alpha}(\sigma)$ resembles the anelastic approximation in Cartesian $z$-coordinates. However, while the continuity equation for the anelastic approximation reduces to that for incompressible flow, the continuity equation in $\sigma$-coordinates is unchanged by this approximation. As a result of this difference, (barotropic) acoustic waves are still supported, while they are not with the anelastic approximation.

As a result of the homogeneous density approximation, the pressure gradient terms become linear, because $\overline{\alpha}\sigma$ only varies with height. In fact, using the ideal gas law $\overline{\alpha}\sigma = \frac{R\overline{T}}{p_s}$, the product $\overline{\alpha}\sigma$ becomes constant with an isothermal background state. Further, vertical temperature advection is approximated by advection of the homogeneous background state only, $\dot{\sigma}\frac{\partial T}{\partial \sigma} \approx \dot{\sigma}\overline{\Gamma}$ with $\overline{\Gamma} = \frac{\partial \overline{T}}{\partial \sigma}$.

In summary, the resulting momentum and thermodynamic equations are

$$\frac{du}{dt} - (f_0 + \beta y)v = -\frac{\partial \phi}{\partial x} - \overline{\alpha}\sigma\frac{\partial p_s}{\partial x} - ru + D\nabla^2 u \text{ ,} \tag{10}$$

$$\frac{dv}{dt} + (f_0 + \beta y)u = -\frac{\partial \phi}{\partial y} - \overline{\alpha}\sigma\frac{\partial p_s}{\partial y} - rv + D\nabla^2 v \text{ ,} \tag{11}$$

$$\frac{d_h T}{dt} + \overline{\Gamma}\dot{\sigma} - \frac{\overline{\alpha}}{c_p}\omega = \frac{J}{c_p} \text{ .} \tag{12}$$

Continuity (eq. 6), hydrostasy (eq. 4) and eqs. (7)-(9) that determine the surface pressure tendency and the vertical velocities all remain unchanged.

### 2.1.3   Quasi-geostrophy

In QG, the geostrophic wind follows from the geostrophic streamfunction $\psi_g$ ,

$$\boldsymbol{v}_g = -\boldsymbol{k} \times \nabla \psi_g \text{ ,} \tag{13}$$

which is in turn defined by

$$f_0\psi_g = \phi + \overline{\alpha}\sigma p_s \text{ .} \tag{14}$$

The prognostic equations are

$$\frac{d_g T}{dt} + \overline{\Gamma}\dot{\sigma} - \frac{\overline{\alpha}}{c_p}\omega = \frac{J}{c_p} \text{ and} \tag{15}$$

$$\frac{d_g p_s}{dt} = \frac{\partial p_s}{\partial t} + \boldsymbol{v}_{gs} \cdot \nabla p_s = \omega_s \,, \tag{16}$$

in which the surface pressure tendency equation (16) is evaluated at the lower surface ($\sigma = 1$), indicated by the subscripts $s$.

The remaining unknown variables at this point are the vertical wind $\dot{\sigma}$ and the pressure vertical velocity $\omega$. Starting from the following form of continuity

$$\nabla_\sigma \cdot \boldsymbol{v}_a + \frac{1}{p_0}\frac{\partial \omega}{\partial \sigma} = 0 \,, \tag{17}$$

the pressure vertical velocity can be derived from the $\omega$-equation

$$\overline{s}\nabla_\sigma^2 \omega + \frac{f_0^2}{p_0}\frac{\partial^2 \omega}{\partial \sigma^2} = 2\nabla_\sigma \cdot (Q_1, Q_2) + \beta f_0 \frac{\partial v_g}{\partial \sigma} + f_0 \frac{\partial r\zeta_g}{\partial \sigma} - f_0 \frac{\partial D\nabla^2 \zeta_g}{\partial \sigma} - \frac{\kappa}{\sigma}\nabla^2 J \,, \tag{18}$$

in which $Q_1$ and $Q_2$ represent the two components of the Hoskins et al. (1978) Q-vector

$$Q_1 = +f_0 \frac{\partial \boldsymbol{v}_g}{\partial x} \cdot \nabla_\sigma \frac{\partial \psi_g}{\partial \sigma} \,, \tag{19}$$

$$Q_2 = -f_0 \frac{\partial \boldsymbol{v}_g}{\partial y} \cdot \nabla_\sigma \frac{\partial \psi_g}{\partial \sigma} \,. \tag{20}$$

Further, $\overline{s} = \overline{s}(\sigma)$ is a horizontally homogeneous stability parameter

$$\overline{s} = -\frac{\overline{\alpha}}{\sigma}\frac{c_v}{c_p} - \frac{\partial \overline{\alpha}}{\partial \sigma} \,. \tag{21}$$

In order to solve the elliptic $\omega$-equation (18), we require a condition for $\omega$ at the lower surface. We obtain $\omega_s$ from the vorticity tendency equation evaluated at the lower surface. The resulting condition is

$$\overline{\alpha}_s \nabla^2 \omega_s - \frac{f_0^2}{p_0}\frac{\partial \omega}{\partial \sigma}\bigg|_s = -\boldsymbol{v}_{gs} \cdot \nabla(\nabla^2 \phi_s) - \beta f_0 v_{gs} - f_0 r_s \zeta_{gs} + f_0 D_s \nabla^2 \zeta_{gs} \,, \tag{22}$$

in which the subscript $s$ again indicates variables that are evaluated at the surface. Finally, after using $\omega_s$ to obtain $\omega$ for the entire model domain, we derive $\dot{\sigma}$ from

$$\dot{\sigma} = \frac{1}{p_0}(\omega - \sigma\omega_s - \sigma(\boldsymbol{v}_g - \boldsymbol{v}_{gs}) \cdot \nabla p_s) \,, \tag{23}$$

which closes the system of equations.

QG potential vorticity $q$ is not used anywhere in the model. Nevertheless, for reference, $q$ takes the form

$$q = \nabla_\sigma^2 \psi_g + \beta y + \frac{f_0^2}{p_0}\frac{\partial}{\partial \sigma}\left[\frac{1}{\overline{s}}\frac{\partial \psi_g}{\partial \sigma}\right] \tag{24}$$

in this QG system.

## 2.2 Ocean dynamics

In addition to the atmospheric component, Bedymo also includes a slab ocean. The slab ocean is intended to represent an oceanic mixed layer that interacts with the atmosphere on time scales on which the internal ocean dynamics can be neglected. Nevertheless, as the oceanic heat transport might play a role even at these time scales, we provide several options to provide oceanic flow and heat transport.

The only prognostic variable in the slab ocean model is the mixed layer temperature $T^o$,

$$\frac{\partial T^o}{\partial t} = -\frac{F_{sh}}{\rho^o c_p^o H} - OHT - \alpha(T^o - T_e^o). \tag{25}$$

It can change due to sensible heat exchange $F_{sh}$ between the ocean and the atmosphere and oceanic heat transport $OHT$. In addition, the model includes a relaxation term towards a prescribed climatological temperature $T_e^o$, which may be used to crudely represent the neglected oceanic circulation (see Tab. 2 for other symbols).

The heat exchange is parameterised with a bulk flux formulation,

$$F_{sh} = \rho_s^a c_p^a C_{sh} |v_s^a| (T_s^a - T^o). \tag{26}$$

Here and in the following, atmospheric variables are denoted by a superscript $a$, and the index $s$ denotes values at the interface between the atmosphere and ocean. Details on how these surface variables are defined are given in section 2.4.

The options for parametrising the oceanic heat transports are

$$OHT = \begin{cases} 0 & \text{for a 0.5-layer model,} \\ v^o \cdot \nabla_h T^o & \text{for a 1-layer model, and} \\ \nabla \cdot (u^o T^o) & \text{for a 1.25-layer model.} \end{cases} \tag{27}$$

The first option represents a motionless ocean, while the other two options differ in the way divergence in the flow field is treated, with $u^o$ denoting the 3D-flow field and $v^o$ the horizontal part of the flow. Whereas divergence does not influence the mixed layer temperature in the 1-layer model, the 1.25-layer model assumes a compensating vertical flow (positive upwards)

$$w^o = \nabla_h \cdot v^o$$

that transports water from a lower layer of temperature $T_2^o$ into the mixed layer,

$$\nabla \cdot (u^o T^o) = \nabla_h \cdot (v^o T^o) - \begin{cases} w^o T^o & \text{if downwelling, } w^o < 0, \\ w^o T_2^o & \text{if upwelling, } w^o \geq 0. \end{cases}$$

The temperature $T_2^o$ might vary spatially, but is kept constant over time. By not letting $T_2^o$ vary with time, we implicitly assume the deeper layer to be motionless and to have infinite heat capacity.

Formally, both the 1-layer model and the 1.25-layer model for the $OHT$ do not conserve energy, because the average mixed layer temperature can change without temperature relaxation or heat exchange with the atmosphere. Upwelling in the 1.25-layer model lets the average mixed layer temperature approach the deep layer temperature, without the latter changing due to

the downwelling required by continuity. The 1.25-layer model reduces to the 1.0-layer model if $T_2^o = T^o$, such that the average mixed layer temperature will over time approach the local mixed layer temperature in locations of divergent flow.

The total transporting flow $v^o = v_e + v_p$ is a combination of a user-prescribed flow pattern $v_p$ and the wind-driven Ekman-flow $v_e$. Our formulation of $v_e$ also includes a small parameter $\epsilon$ in addition to the standard Ekman flow,

$$v_e = \frac{\rho_s^a C_D |v_s^a|}{\rho^o H(\epsilon^2 + f^2)} (\epsilon v_s^a - f k \times v_s^a), \tag{28}$$

which controls the magnitude of the non-rotating flow excited by wind stresses. This addition results in finite Ekman velocities at the equator. Codron (2012) refers to $\epsilon$ as the inverse damping timescale of oceanic currents.

### 2.3 Numerics and model infrastructure

#### 2.3.1 Coordinate system and discretisation

Bedymo uses Cartesian coordinates in the horizontal and a terrain-following pressure coordinate $\sigma$ in the vertical, which is defined by

$$\sigma = \frac{p}{p_s} . \tag{29}$$

Variations in surface pressure thus affect the coordinate system throughout the atmospheric column up to the model top at $p = 0$.

In both the horizontal and the vertical directions, the velocity components are staggered with respect to the main grid points, following the C-grid setup of Arakawa and Lamb (1977). Temperature, geopotential, specific volume, and surface pressure are all defined on the unstaggered grid.

Bedymo uses the $3^{\text{rd}}$-order Runge-Kutta time integration scheme in conjunction with a $3^{\text{rd}}$-order upwind-biased interpolation of the advected quantities to determine the advective fluxes at the staggered locations of the wind speed components, following the derivations of Smolarkiewicz (1982) and Tremback et al. (1987). Only for vertical advection in the uppermost and lowermost level a $1^{\text{st}}$-order upwind-biased interpolation is used. We tested higher and lower order advection schemes for the interior of the model domain, but this choice turned out to be largely inconsequential. The explicit scale-selective damping active for most of the test setups is dominating the numerical diffusion (cf. Tab. 3). The remainder of the model is discretised using $2^{\text{nd}}$-order centred differences and interpolations.

#### 2.3.2 Elliptical solver

In QG-mode, an elliptic equation has to be solved to determine the vertical velocity. Through extensive testing, we found the Full Multigrid Method (e.g. Saad, 2003) in conjunction with the Bi-Conjugate gradient Stabilised (BiCGstab) Method of van der Vorst (1992) to be the most effective and stable configuration for Bedymo. We use BiCGstab both to solve the elliptic equation on the coarsest grid and to iteratively refine the solution on finer grids.

## 2.4 Boundary conditions

The boundaries of the model domain are located at staggered grid points. At the model top, $p = 0$ and also both vertical velocities vanish, $\dot{\sigma} = 0$ and $\omega = 0$. At the surface, $\dot{\sigma} = 0$, but $\omega_s \neq 0$ as surface pressure can change both locally and in a Lagrangian reference frame. In QG, $\omega_s$ is determined through a condition derived from the surface vorticity tendency (eq. 22); in PE, it is derived from continuity (eq. 9).

There are two available options for boundary conditions along the lateral boundaries. The first option represents periodic boundaries in the respective direction, and the second option features an impermeable free-slip "wall" with zero boundary-normal fluxes.

## 2.5 Python bindings and graphical user interface

While the model is written in Fortran, the code is structured such to make it easily accessible from python using F2PY (NumPy Developers, 2022). In particular, the model can be run from python and model variables are accessible from python during runtime.

These python bindings are the basis for a rudimentary python run script illustrating the basic use of the model through python, as well as a graphical user interface (GUI) based on python and PyQt (Fig. 1). The GUI allows to start, stop, and single-step the model and watch the evolution of all prognostic as well as pertinent diagnostic variables interactively while the model is running. Both the run script and the GUI are included in the Bedymo source code repository.

## 3 Evaluation

We subject Bedymo to several test cases to evaluate the performance of the model. We chose different test cases to isolate pertinent aspects of the atmospheric and coupled dynamics. Furthermore, we chose test cases that have been studied comprehensively, such that the expected results are well established. The first four test cases focus on important aspects of the mid-latitude atmospheric dynamics in isolation. In a fifth test case, we evaluate the PE model in conjunction with the slab ocean by simulating the coupled response to a temperature anomaly in the ocean mixed layer located at the equator. A summary of pertinent model parameters for all test cases is given in Tab. 3.

### 3.1 Atmospheric test cases

The first atmosphere-only test case evaluates the representation of baroclinic cyclogenesis in a mid-latitude channel on the $f$ and $\beta$-plane, as well as the sensitivity of the baroclinic development to the magnitude of the initial baroclinicity. Second, we extend the baroclinic channel setup by including temperature relaxation and friction and evaluate the long-term zonal-mean statistics of the storm track simulated by Bedymo. Finally, test cases three and four evaluate the development towards the stationary wave responses to isolated orography, with a focus on Rossby waves and inertia-gravity waves, respectively.

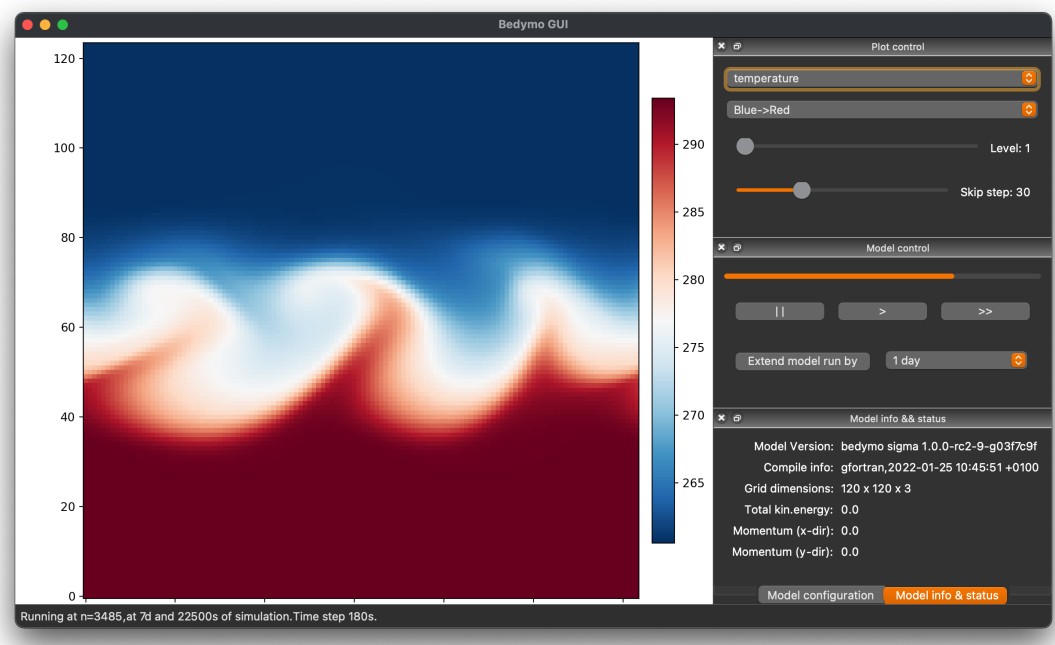

**Figure 1.** Screenshot of the graphical user interface for Bedymo taken while the model is running.

### 3.1.1 Baroclinic cyclogenesis on an $f$ and $\beta$-plane

The baroclinic channel for the cyclogenesis test case is 16000 km long and periodic in the zonal direction. In the meridional direction it is 12000 km wide and bounded by impermeable walls. The horizontal resolution is 100 km and we use three levels in the vertical. We initialise a baroclinic zone with a temperature contrast of in total about 32 K distributed over about 3000 km. Temperature stratification is determined by $\overline{\Gamma} = 0.8\overline{\Gamma}_d$, with $\overline{\Gamma}_d$ the dry adiabatic lapse rate in sigma-coordinates. This corresponds approximately to an overall Brunt Vaisala frequency $N^2 = 1.2 \cdot 10^{-4}\,\mathrm{s}^{-2}$. The baroclinic zone is meridionally

centred in the channel. There is no initial surface pressure gradient and thus no initial surface winds. With Coriolis parameters corresponding to a latitude of 50°N, the baroclinic zone is initially balanced by a jet of maximum intensity of about 50 m s$^{-1}$ in the uppermost layer ($\sigma = 1/6$).

To localise the baroclinic development, we perturb this initial state by a warm anomaly of 2 K located at the center of the baroclinic zone. The temperature perturbation is invariant with height and has a zonal and meridional extent corresponding to

the width of the baroclinic zone (approx. 3000 km). The perturbation is initially balanced by a perturbation thermal wind.

In our control setup, we use the $\beta$-plane approximation, as it is this setup we will later extend to multi-decadal storm track simulations. We will compare this control setup to simulations on an $f$-plane and to simulations in which we vary the initial baroclinicity to yield a balanced initial jet of either about 30 m s$^{-1}$ or 70 m s$^{-1}$, respectively.

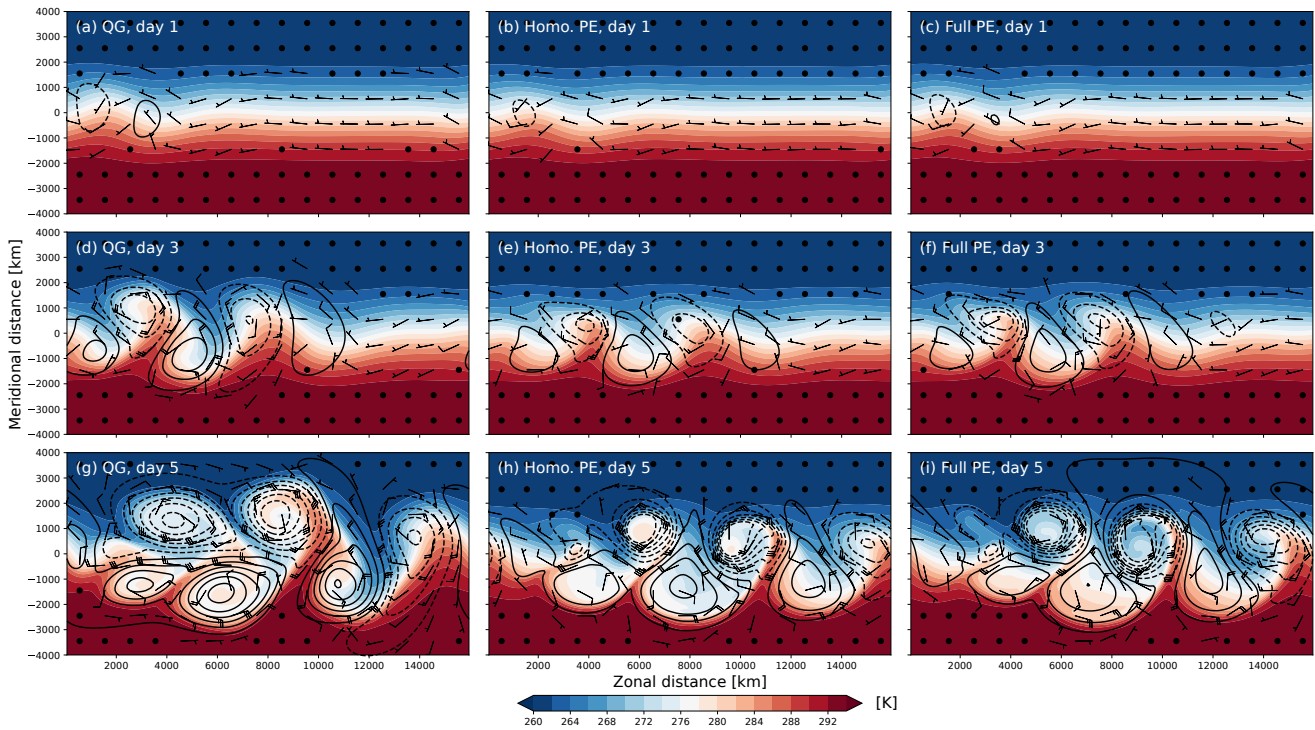

**Figure 2.** Baroclinic development in a three-layer channel model on an $\beta$-plane for (a,d,g) QG and (b,e,h) homogeneous density PE, and (c,f,i) full PE. From top to bottom, the rows show the development after (a-c) 1 days, (d-f) 3 days, and (g-i) 5 days lead time, respectively. The shading shows temperature in Kelvin and barbs the winds in $\mathrm{m\,s^{-1}}$, both representing the lowest of the three layers at $\sigma = 5/6$. The contours show surface pressure with a contour interval of 10 hPa centred on 1000 hPa, which contours below 1000 hPa dashed. Note that the not the entire model domain is shown in the $y$-direction.

The baroclinic development for the control setup and the three model variants is summarised in Figure 2. For the first three
235   days, the evolution in the three model variants is quite similar (Fig. 2a-f), and only at day 5 structural differences become apparent between QG and the two PE variants (Fig. 2g-i). In QG, cyclones are considerably larger in scale than in the PE variants and largely symmetric in size and structure compared to the anticyclones. In contrast, PE cyclones are smaller, more circular, and surrounded by steeper pressure gradients than PE anticyclones. Further, only PE produces qualitatively realistic fronts with near-discontinuities in the temperature field. This is expected from theory, as Hoskins (1975) showed that advection
by ageostrophic winds must be taken into account in order to realistically capture the evolution of fronts.

Comparing the two PE variants, only minor differences occur up to day 5 (Fig. 2h-i). The two most apparent differences are a slightly slower cyclone intensification with homogeneous density PE, and, potentially related, changes in the temperature structure. In the homogeneous density PE, cyclone cores are comparatively warmer, and anticyclones comparatively cooler than their counterparts in full PE. Given that the only differences between the two PE variants is linearised vertical advection

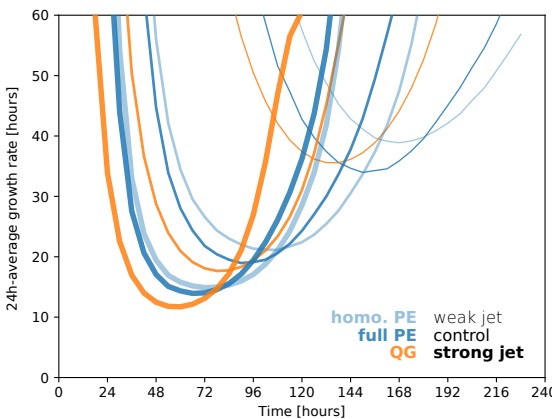

**Figure 3.** Growth rate of eddy kinetic energy averaged over 24-hour periods for QG (orange) and the two variants of PE (blue, light blue). Line weight indicates the initial and jet strength with maximum winds of $30\,\mathrm{m\,s^{-1}}$ (thin), $50\,\mathrm{m\,s^{-1}}$ (medium), and $70\,\mathrm{m\,s^{-1}}$ (bold).

and a linearised pressure gradient force, it seems implausible that these differences in temperature originate from the linearised vertical advection. Warmer cyclones and colder anticyclones yield an overall somewhat reduced baroclinicity in homogeneous density PE compared to full PE, which is consistent with the slightly slower intensification in homogeneous density PE.

Considering the growth rate of the eddy kinetic energy, both variants of PE develop somewhat slower and later than QG (Fig. 3), with maximum growth rates of 17.7 hours (QG), 19.0 hours (full PE), and 21.1 hours (homo. PE). For our channel

setup, the Rossby deformation radius is approximately $930\,\mathrm{km}$ (e.g. Vallis, 2006), which implies a most unstable wave length of about $3600\,\mathrm{km}$, and a maximum Eady growth rate of

$$\sigma_E = 0.31 \frac{u_{max}}{L_d} \approx 1.66 \cdot 10^{-5}\mathrm{s^{-1}},$$

for the control jet intensity of $50\,\mathrm{m\,s^{-1}}$. This corresponds to an e-folding time scale of $\tau_E = 16.7\,$hours, which fits quite well, in particular with the QG variant.

Further, the simulated evolution generally follows the expectations from theory for varying magnitudes of the baroclinicity. For a stronger jet of $70\,\mathrm{m\,s^{-1}}$, the Eady growth time scale reduces to $11.9\,$hours. This fits almost perfectly to the simulated maximum intensification with a time scale of $11.7\,$hours in the QG variant (darkest blue line in Fig. 3). The two PE variants are again somewhat slower and later in their development (lighter blue lines in Fig. 3). For the weaker jet of $30\,\mathrm{m\,s^{-1}}$, the correspondence between theory and simulation is less tight, with an Eady growth time scale of $27.8\,$hours and a QG maximum

intensification with a time scale of $35.6\,$hours. This might, at least partly, be due to the beta-effect and downstream development playing a comparatively larger role with weaker baroclinicity. But this cannot be the full explanation, because for full PE the reduction in the peak growth rate (about 44%) is only slightly larger than the reduction in baroclinicity (40%).

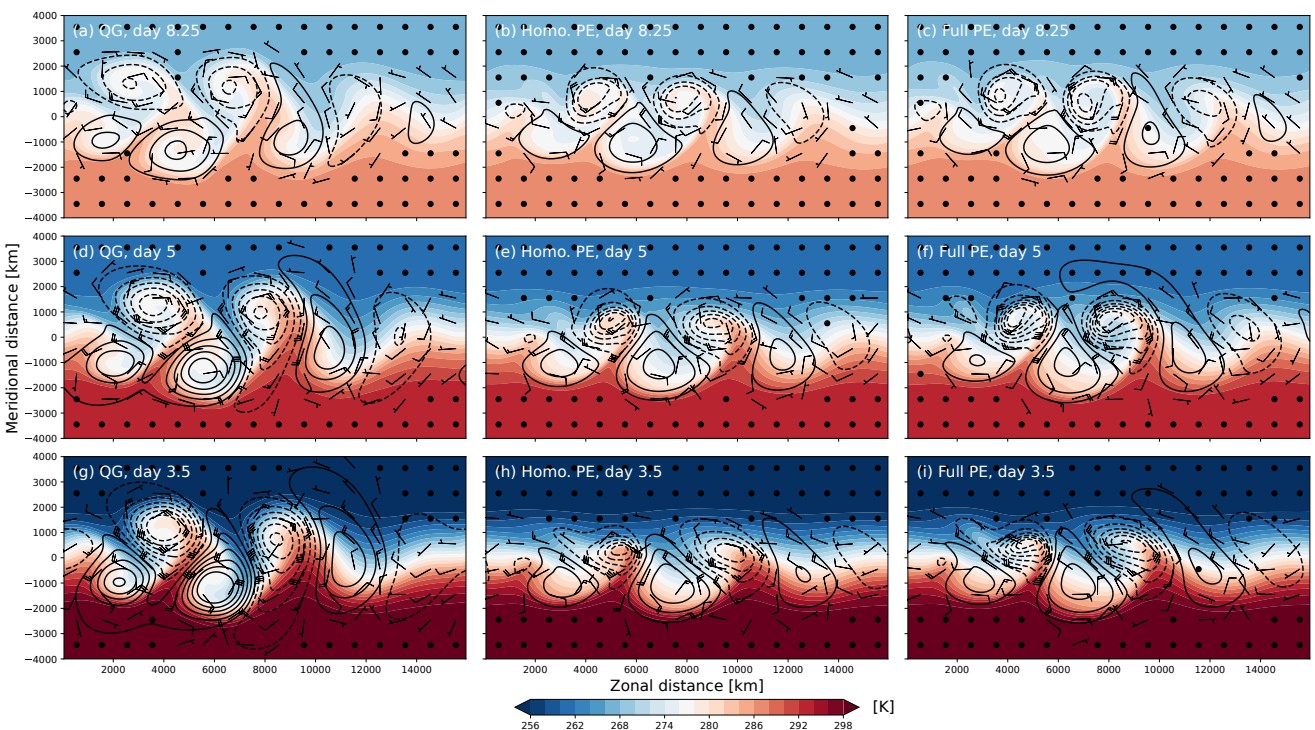

**Figure 4.** Sensitivity of the downstream development to the initial jet speed in a three-layer channel model on an $\beta$-plane using jet speeds of (a-c) $30\,\mathrm{m\,s}^{-1}$, (d-f) $50\,\mathrm{m\,s}^{-1}$, and (g-i) $70\,\mathrm{m\,s}^{-1}$. The presented lead times are scaled linearly with the initialised jet speed, such that the rows show the development after (a-c) $8.25\,\mathrm{days}$, (d-f) $5\,\mathrm{days}$, and (g-i) $3.5\,\mathrm{days}$ lead time, respectively. Columns show the different versions of the model; panels (a,d,g) show QG, (b,e,h) homogeneous density PE, and (c,f,i) full PE. Barbs and contours as in Fig. 2, and panels (d-f) identical to panels (g-i) of Fig. 2 except for the temperature scale.

Despite these differences in growth rate, structurally the evolution remains very consistent across the three tested magnitudes of the baroclinicity (Fig. 4). Considering lead times normalised by the different jet intensities, the wave structures are nearly identical within each model variant (columns in Fig. 4), showing that downstream propagation of wave energy scales largely linearly with the jet intensity despite the considerable amplitude of the waves.

The evolution of the cyclone is qualitatively similar on the f and $\beta$-plane for all three model variants (compare Figs. 2 & A1). In particular the differences in the size and shape of cyclones and anticyclones translate from the $\beta$ to the f-plane. However, at day 5 it becomes apparent that meridional movement is much less constrained on the f-plane compared to the $\beta$-plane. The meridional scale of the synoptic systems is markedly larger, and soon after day 5 the large cold sectors in the zonal center of the domain start to interact with the southern boundary of the domain.

Overall, the structure and sensitivities of the simulated cyclone development is very much in line with previous simulations of idealised cyclones with more complex models (e.g., Schemm et al., 2013, Terpstra and Spengler, 2015, and the cyclones in

the colder environments of Tierney et al., 2018). This consistency is not surprising given the much earlier results of Simmons and Hoskins (1978), for instance, which used a model similar in complexity to Bedymo to study cyclogenesis. But the consistency remains worth noting, because these more recent studies typically employ a factor of 5-10 higher resolution in both the horizontal and vertical and use a full suite of physics parametrisations.

### 3.1.2 Mid-latitude storm track

The control setup for the baroclinic instability test case also serves as the basis for long-term simulations to evaluate the representation of a mid-latitude storm track in Bedymo. In order to achieve a statistically stationary storm track, we follow Held and Suarez (1994) and add simple parametrisations for three physical processes to the model setup. First, we enable a temperature relaxation towards the initial state with a time scale of $10^6$ s $\approx 11.6$ days throughout the model domain to represent all process that replenish baroclinicity in real storm tracks. Second, we enable linear Ekman friction in the lowest model layer with $r = 10^6$ s. Finally, we enable bi-harmonic diffusion that damps $2\Delta x$-waves with a time scale of $10^5$ s $\approx 28$ hours. This time scale increases quadratically with increasing wave length and thus predominantly affects the shortest waves. The storm track simulations cover a time period of 32 years of 360 days each, of which the first two years are discarded as spin-up. After 1-1.5 years there is no discernible trend anymore in the distribution of sea-level pressure in any of the simulations, which we take as an indication that a statistical equilibrium has been reached.

Because of the absence of any zonal asymmetries, the resulting storm track statistics are zonally symmetric. Figure 5 thus presents the zonal and time average state for the last 30 years of the simulations. In addition to the average temperature and zonal winds, we also show the eddy momentum and heat fluxes, in which the temperature and wind perturbations (indicated by primes) are taken to be the deviations from the zonal-and-time average. These fluxes are interesting, because, for example $\frac{\partial}{\partial y}\overline{u'v'} < 0$ indicates a convergence of eddy momentum fluxes in the time mean, and analogously for $\frac{\partial}{\partial y}\overline{v'T'} < 0$ and $\frac{\partial}{\partial \sigma}\overline{\dot{\sigma}'T'} < 0$ that indicate a convergence of heat in the meridional and vertical, respectively. To keep the line plots legible, we restrict our presentation and discussion to QG and full PE.

In QG, the resulting storm track is symmetric and centred on the imposed baroclinic zone. In the angular momentum budget, weak surface easterlies on either side of the baroclinic zone balance the near-surface westerlies under the jet (Fig. 5a). Maximum zonal winds in the uppermost level are just below $50\,\mathrm{m\,s^{-1}}$, which would be the wind speed required to thermally balance the initial and relaxation temperature gradient in the absence of near-surface winds. Consistent with the mean zonal winds, the mean convergence of momentum is symmetric around the imposed baroclinic zone with maximum momentum convergence at the jet core which is also strongest at the jet level (Fig. 5b). In contrast, heat transports are strongest in the lowest level (Fig. 5d, e), consistent with the largest deviation of mean temperature from the relaxation state (Fig. 5c).

The described QG storm track is qualitatively similar to the storm tracks produced by other models (e.g., Held and Suarez, 1994; Vallis et al., 2004; Frierson et al., 2006; Voigt and Shaw, 2016) and the ones actually observed (e.g., Figs. 7.7 and 11.8 of Peixoto and Oort, 1992). While these qualitative similarities are encouraging, the evaluation of Bedymo through comparison with existing models is somewhat limited by important differences between these models. Aquaplanet and Held-Suarez-type model simulations are global and thus both include a Hadley circulation and take into account the spherical geometry of the

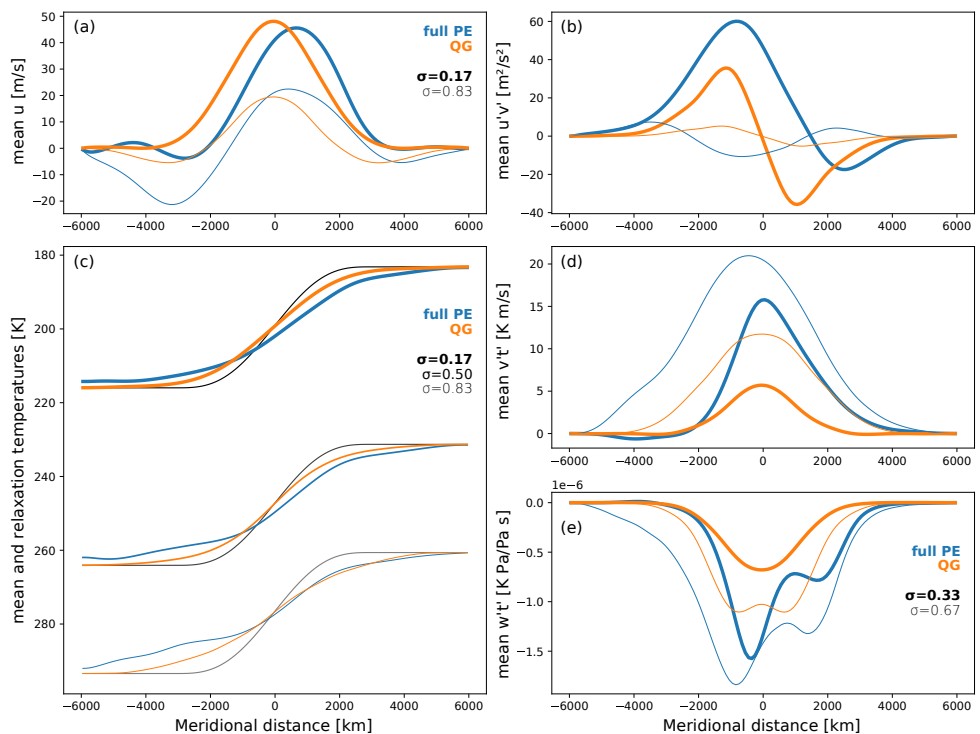

**Figure 5.** Statistical equilibrium state of the QG and full PE storm tracks, respectively. Line colour and weight are consistent throughout all the panels and identical for panels (a), (b) and (d). Throughout, blue lines represent full PE, and orange lines QG, respectively. Line weight indicates the vertical level, with the boldest lines representing the uppermost level.

Earth. Both of these features are missing in Bedymo but will certainly affect the simulated storm track in these other models. For example, by construction, the storm track and jets in Bedymo are purely eddy-driven, while Held-Suarez-type simulations also include a thermally driven subtropical jet.

In contrast to QG, the simulated PE storm track is not symmetric around the imposed baroclinic zone. The core of the jet is shifted poleward at all levels, and near-surface easterlies mainly occur on the equatorward side of the imposed baroclinic zone. This asymmetry is likely related to the asymmetry between cyclones and anticyclones observed in the cyclogenesis test case (Fig. 2). As discussed there, the strongest gradients in both sea-level pressure and temperature occur around cyclones, and thus on average somewhat poleward of the imposed baroclinic zone (Fig. 2). It thus seems plausible that the time mean reflects this asymmetry. If this interpretation is correct, the asymmetric storm track would thus be another consequence of taking into account advection by the ageostrophic winds (cf., Hoskins, 1975; Wolf and Wirth, 2015).

The storm tracks observed on Earth display a similar meridional asymmetry with regards to the surface winds. Near-surface easterlies are much more pronounced in the subtropics, on the equatorward side of the baroclinic zone, compared to the polar regions. Despite the similarity, the mechanisms leading to these easterlies might nevertheless be different, with spherical

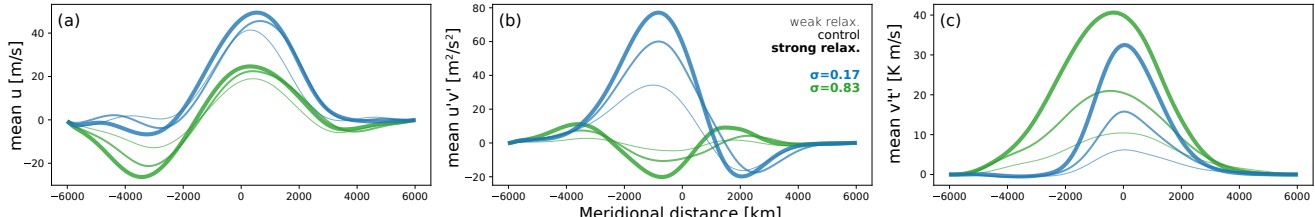

**Figure 6.** Sensitivity of the full PE storm track to the strength of temperature relaxation. The panels (a-c) here correspond to panels (a), (b) and (d) in Fig. 5, with the control simulation here being identical to the PE simulation there. Line colour and weight is consistent across panels.

geometry and the Hadley circulation again likely affecting the dynamics. In fact, some setups of the spherical 1-layer QG model of Vallis et al. (2004) yield meridional asymmetries similar to the one seen here for PE.

In addition to the asymmetries, the PE storm track features larger heat and momentum fluxes at all levels compared to QG. Nevertheless, the meridional profiles of these fluxes are generally consistent across QG and PE at all levels. The only exception
is near-surface eddy momentum transport, but here amplitudes are quite small both in QG and PE. Consistent with the more vigorous heat transport in PE, the average temperature deviations from the relaxation state are larger in PE than in QG. On the equatorward side of the mean jet, weak but noticeable temperature deviations extend all the way to the southern boundary. Interactions with the boundary do, however, not affect the PE storm track, as the meridional profiles of all parameters remain nearly unchanged in a simulation with a meridionally wider channel.

Finally, the simulated storm track remains largely consistent across a range of temperature relaxation time scales (Fig. 6). In addition to the default relaxation with a time scale of 11.6 days, we conducted simulations with full PE using time scales of 3.47 and 34.7 days, respectively. These simulations are called "strong" and "weak" in Fig. 6, respectively. Within this range of parameters, stronger relaxation yields a more vigorous storm track with more intense heat and momentum transports as well as stronger jets (Fig. 6).

### 335 3.1.3 Rossby waves excited by orography

The third mid-latitude test case evaluates the developing stationary wave response to isolated orography. Orographically forced stationary waves are one of the main ingredients determining the zonal asymmetries in the Northern Hemisphere storm tracks (Held et al., 2002). The test case considered here uses the same mid-latitude channel used in the previous test cases, but the model is initialised by homogeneous zonal winds of $10\,\mathrm{m\,s^{-1}}$ which are balanced by a surface pressure gradient. In order to
340 better resolve the vertical structure of the Rossby wave pattern, we use 10 vertical levels compared to 3 for the storm track test cases. Orography is introduced as an isolated elliptical cone-shaped mountain, meridionally centred in the domain, with a meridional scale of 1500 km, a zonal scale of 1000 km, and a height of $100\,\mathrm{m^2\,s^{-2}}$.

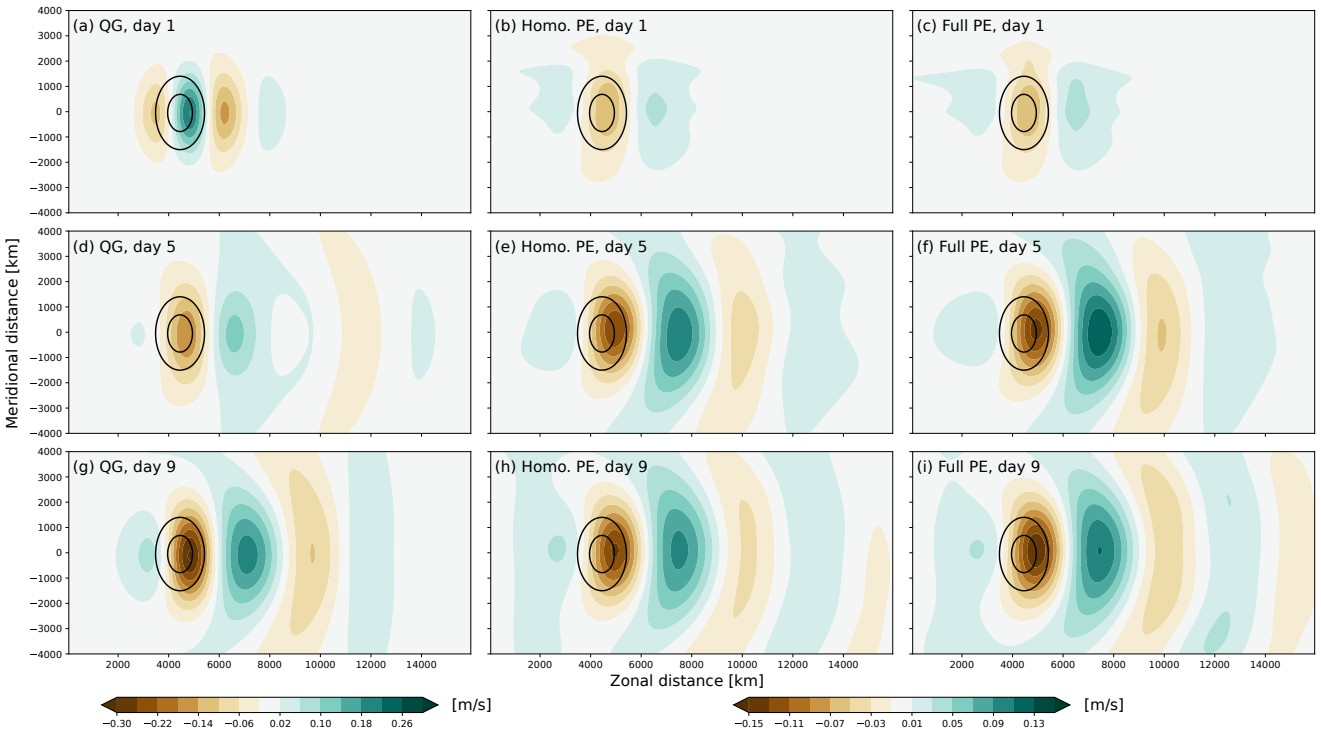

**Figure 7.** Developing stationary Rossby wave response to orographic forcing on a $\beta$-plane for (a,d,g) QG and (b,e,h) homogeneous density PE, and (c,f,i) full PE. From top to bottom, the rows show the development after (a-c) 1 days, (d-f) 5 days, and (g-i) 9 days lead time, respectively. The shading shows the meridional wind at $\sigma = 0.75$ in m s$^{-1}$. The contours show the orography by the 4 m and 8 m-contours. Note that not the entire model domain is shown in the $y$-direction.

Finally, we increase the scale-selective damping parameter $D$ to $1 \cdot 10^{-4}$ m$^2$ s$^{-1}$ to be able to damp out gravity waves from slight initial imbalances in the PE solutions. These imbalances are zonally symmetric thus appear as zonally symmetric anomalies in the meridional wind on top of the Rossby wave pattern. Such anomalies are visible at day 1 and 5 (Fig. 7b,c,e,f), but have largely disappeared at day 9 (Fig. 7h,i).

Figure 7 shows the developing Rossby wave train excited by the mountain for the three model variants. As expected from theory, the QG response is meridionally symmetric. In contrast, the two variants of the PE again show slight meridional asymmetries. These asymmetries become increasingly more pronounced with increasing mountain height (not shown), and are thus most likely due to non-linear interactions of the perturbation flow with itself. But despite the slight asymmetries in the PE solutions, the response is very consistent in both shape and amplitude across the three model variants.

Further, the response fits qualitatively well to the response expected from linear wave theory (Fig. 8; Hoskins and Karoly, 1981; Held et al., 2002). In particular the horizontal wave pattern compares very well between the analytic model and Bedymo, whereas there is about a factor 2 or 4 difference in amplitude compared with the PE and the QG solutions, respectively (Figs. 7

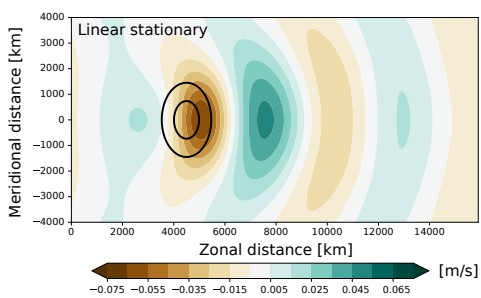

**Figure 8.** Linear quasi-geostrophic solution of the stationary Rossby wave response to isolated orography. Shading and contours as in Fig. 7 except that the meridional wind is shown at $z = 2500\,\text{m}$.

& 8). For reasonably low values of the damping parameter (corresponding to $r$ in Bedymo), the amplitude in the analytic model the amplitude is first and foremost set by the lower boundary condition, which is markedly different from both the QG and the PE modes in Bedymo. In the analytic model, the lower boundary is given by a temperature anomaly at $z = 0$, whereas in Bedymo it is given by a surface geopotential at $\sigma = 0$. The surface geopotential enters the QG system by modifying the surface vorticity field (cf. eq. 22) whereas it enters the PE system through the continuity equation. Thus, only in the PE mode

of Bedymo does the mountain actually block any volume. Some discrepancies with respect to the amplitude of the wave pattern are thus to be expected.

### 3.1.4 Inertia-gravity waves excited by orography

The fourth and final mid-latitude test case evaluates the developing stationary inertia-gravity wave (IGW) pattern response to isolated orography. In comparison to the previous Rossby-wave test case, all horizontal scales, including the extent of

365 the mountain and the grid spacing, are decreased by an order of magnitude. We further increased the number of vertical levels to 30 in order to properly resolve the vertical propagation of the IGWs. To isolate the IGW from the Rossby-wave response, we approximate the Coriolis effect by an $f$-plane. $QG$ cannot represent IGWs, and the two variants of PE yield nearly indistinguishable results. We thus focus our presentation on the results from full PE.

Again we compare the transient solutions from Bedymo (Fig. 9) with the linear stationary solution (Fig. 10). Both the

370 horizontal and the vertical wave propagation patterns fit qualitatively very well. Only the wave amplitude is again about a factor 2 larger in the Bedymo-PE solution compared to the linear stationary one. Further, 48 hours into the simulation, the vertical wave patterns shows first signs of interference with waves reflected from the model top (Fig. 9d), whereas the horizontal propagation of the inertia-gravity waves is still largely undisturbed by the model lateral boundaries (Fig. 9c).

### 3.2 Coupled tests

In addition to the mid-latitude test case, we subject Bedymo to a test case centred on tropical air-sea interaction. The model domain remains a zonally periodic channel with a meridional width of $10000\,\text{km}$. Zonally, the channel is extended to cover the

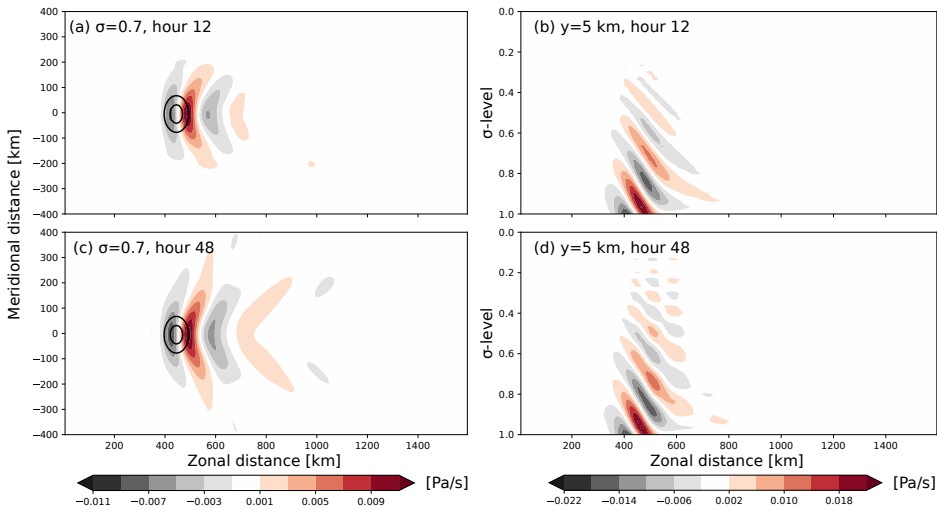

**Figure 9.** Developing stationary inertia-gravity wave response to orographic forcing on an $f$-plane for full PE. The left column (a,c) shows the horizontal wave pattern at $\sigma = 0.7$, the right column (b,d) the vertical wave propagation in the $x$-$z$-plane at $y \approx 0$. From top to bottom, the rows show the development after (a,b) 12 hours and (c,d) 48 hours. The shading shows the pressure vertical wind in $\mathrm{Pa\,s^{-1}}$, the contours show the orography by the 4 m and 8 m-contours. Note that not the entire model domain is shown in the $y$-direction.

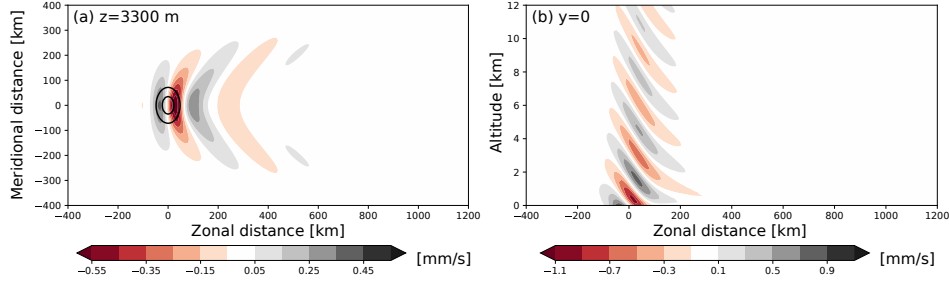

**Figure 10.** Linear stationary inertia-gravity wave response to the forcing from isolated orography. Shading and contours as in each row of Fig. 7 except that the shading shows the Cartesian vertical wind in $\mathrm{mm\,s^{-1}}$.

actual circumference of the Earth of 40000 km and the Coriolis parameter is adapted to represent an equatorial $\beta$-plane. As the model domain contains the equator, we use only the two PE variants of Bedymo for this test case.

Initial surface air temperatures and sea-surface temperatures (SSTs) are homogeneous 288 K except for a $+5$ K SST anomaly with a radius of 1500 km centred on the equator. Below this warm mixed layer is a 15 K colder deeper ocean layer without heat anomaly. In case of the 1.25-layer ocean model, this cold deep water can be upwelled to affect the ocean mixed layer temperature and thus the SST. The atmosphere is initialised with homogeneous and barotropic easterlies with a wind speed of 10 m s$^{-1}$ that is balanced as far as possible and necessary by a meridional surface pressure gradient. Temperature stratification

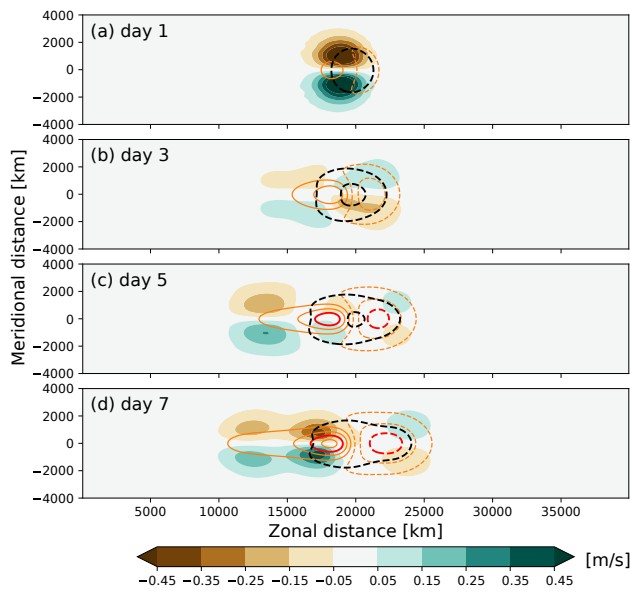

**Figure 11.** Developing wave response in response to an equatorial SST anomaly on a $\beta$-plane for full PE. From top to bottom, the rows show the development after (a) 1 day, (b) 3 days, (c) 5 days, and (d) 7 days lead time, respectively. The panel setup is as in Fig. 7, but showing wind in the lowest of three levels ($\sigma = 5/6$) and with the contour interval for the surface pressure anomaly decreased to 1 hPa. The additional orange-red contours show zonal wind anomalies relative to the initialisation with a contour interval of $1\,\mathrm{m\,s}^{-1}$ centered around zero and the $\pm 2.5\,\mathrm{m\,s}^{-1}$-contour highlighted.

in the atmosphere is as in the cyclogenesis test case, corresponding approximately to $N^2 = 1.2 \cdot 10^{-4}\,\mathrm{s}^{-2}$. Neither ocean nor
atmospheric temperatures are relaxed.

The initial transient atmospheric response to the surface heating is shown in Figure 11. As the ocean heat transport only has a minor influence within the initial seven days shown in Figure 11, and as homogeneous density PE and full PE yield very similar results (cf. Figs. 11 & A2), we here focus on the combination of full PE with the 0.5-layer ocean which ignores ocean heat transport (eq. 27).

Although the forcing from the $+5\,\mathrm{K}$ SST anomaly is clearly finite-amplitude, the initial atmospheric response to equatorial surface heating is consistent with the linear Matsuno-Gill response derived in Matsuno (1966) and Gill (1980). The solution consists of a pair of cyclonic vortices, symmetric on either side of the equator, on the western side of the heating. These vortices represent the Rossby wave-component of the response as derived in Matsuno (1966). These Rossby waves are accompanied by a (baroclinic) Kelvin wave propagating eastwards along the equator Gill (1980). When comparing our transient solution
to those in Gill (1980), it is important to remember that Gill (1980) includes a large amount of damping in order to arrive at a stationary solution. We only apply weak scale-selective damping as described in the storm track test case, and the excited Rossby and Kelvin waves can thus propagate away from the wave source.

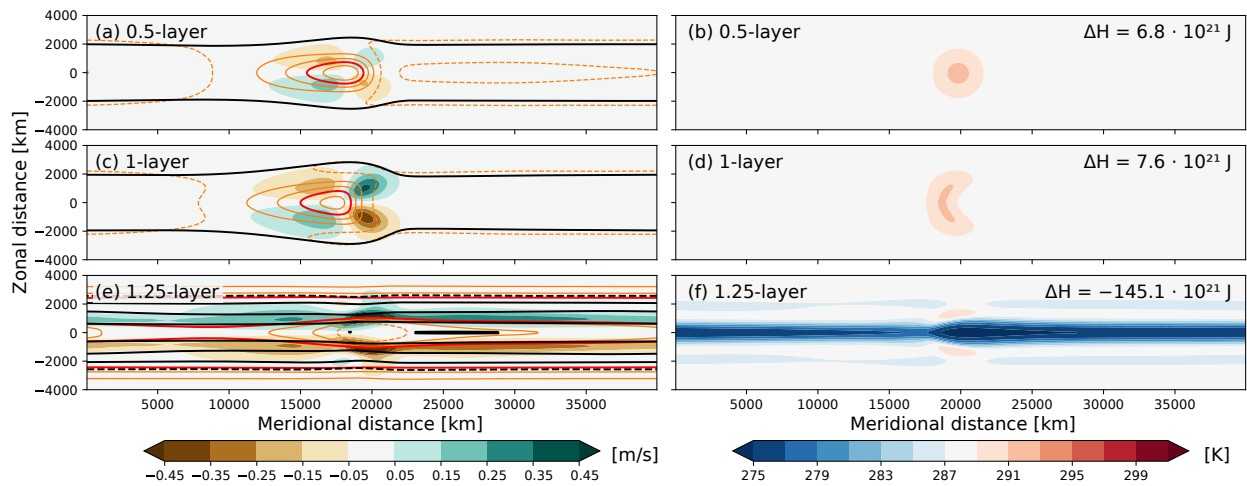

**Figure 12.** Near-stationary wave response to a coupled equatorial SST anomaly on a $\beta$-plane for full PE after 60 days of lead time. From top to bottom, the rows show the development of (a,b) the 0.5-layer model, (c,d) the 1-layer model, and (e,f) the 1.25-layer model, respectively. The panel setup for the left column is as in Fig. 11, but to accommodate the much larger sea-level pressure anomalies in (e), black contours are shown at -1.5 hPa through 1.5 hPa in steps of 1 hPa as well as at 5 and 10 hPa, and zonal wind contours are omitted beyond $2.5\,\mathrm{m\,s^{-1}}$. The maximum zonal wind anomaly on either side of the equator is about $7\,\mathrm{m\,s^{-1}}$. The right column shows ocean mixed-layer temperatures in Kelvin, with the numbers in the top right corners indicating the domain-integral mixed layer heat anomaly relative to 288 K.

After 60 days lead time, the atmosphere achieved a near-equilibrium, which is only slowly evolving in tandem with the evolving ocean. At this time, the coupled solution depends strongly on the chosen parametrisation for the ocean heat transport (Fig. 12). We here again focus the discussion on the simulations using full PE, as the homogeneous density PE solutions are very consistent (cf. Figs. 12 & A3).

With the 0.5-layer ocean, the shape of the initial SST anomaly is entirely intact after 60 days (Fig. 12b), because the ocean itself does not internally redistribute heat. Nevertheless, the ocean mixed layer lost about a quarter of its initial heat anomaly to the atmosphere and the SST anomaly decreased by about 1.2 K. The atmospheric response is much weaker in amplitude than the initial shock-like response to the heating (cf. Figs. 11 & 12a), but it is now following the stationary solution in Gill (1980) more closely, as the transient response has largely dissipated due to numerical and parameterised diffusion.

With the 1.0-layer ocean, direct wind-induced currents and associated ocean heat transports considerably deformed the SST anomaly over 60 days (Fig. 12d). With this parametrisation of the ocean-heat transport, diverging currents do not change the mixed-layer temperature. Atmospheric easterlies induce divergent ocean currents along the equator, thereby increasing the area covered by the warm anomaly without decreasing its amplitude. Overall this leads to a slightly larger ocean heat content with the 1.0-layer compared to the 0.5-layer ocean (compare Fig. 12d and Fig. 12b). Consistent with the slightly larger ocean heat anomaly, the atmospheric response in the 1.0-layer ocean setup is slightly larger in amplitude than for the 0.5-layer ocean (Fig. 12a, c).

Finally, with the 1.25-layer ocean, divergent currents along the equator lead to the upwelling of considerably colder water, leading to large negative SST anomalies along the equator (Fig. 12f). This upwelling is larger in amplitude than the initially positive SST anomaly, which thus largely disappeared after 60 days. In fact, the coldest SSTs appear just eastward of the location of the initial SST anomaly, because the initial response intensified the surface winds in this region. With the warm anomaly largely disappeared, both the SST anomaly field and the atmospheric response is to a first approximation zonally symmetric (Fig. 12e, f). The colder SSTs lead to a marked increase in surface pressure over the equator and a zonally-average meridional flow component away from the equator on either side.

In summary, Bedymo successfully captures both the transient (cf., Matsuno, 1966) and (near-)stationary response (cf., Gill, 1980) to equatorial heating. Beyond these theoretical expectations, the comparison of the different parametrisations of the ocean heat transport demonstrates that both ocean-internal dynamics and air-sea exchange have a profound influence on the coupled solution on time scales longer than a week or so. The atmosphere-ocean interactions occurring in the 1.0-layer and 1.25-layer ocean are however beyond what could be captured by analytic solutions, such that the work of Matsuno (1966) and Gill (1980) can only provide limited guidance. A further caveat with this conclusion is that Bedymo so far cannot represent dynamical balances in the ocean, and thus, for example, cannot represent the ocean gyre circulations in the mid-latitudes.

## 4    Conclusions

We introduced a joint approach to consistently solve the quasi-geostrophic (QG) and two variants of the primitive equations (PE). In all systems, we forecast temperature and surface pressure. In PE, the horizontal wind velocity components also need to be forecasted, whereas all other variables follow diagnostically in QG.

We implemented this approach in the BErgen DYnamic MOdel (Bedymo) and demonstrated the feasibility of the approach as well as the performance of Bedymo on the basis of five test cases. These cases are (a) the baroclinic development of a cyclonic disturbance, (b) the representation of mid-latitude storm tracks, (c,d) the excitation of Rossby and inertia-gravity waves by isolated orography, and (e) the coupled response of the PE models with a slab ocean to an equatorial temperature anomaly in the ocean mixed layer. In all cases, the model results agree well with either an analytical solution for the corresponding linearised problem or conceptual models.

By successfully combining QG and PE into one consistent model, Bedymo considerably simplifies the comparison of the dynamical differences between QG and PE, because it eliminates all error sources associated with comparing two different models. The ability to simply switch between the approximations is especially valuable in cases where the formal validity of QG becomes questionable. One such example is the treatment of orography, because the assumptions underlying QG formally require orographic slopes to be negligibly small.

## 5 Outlook

The development of Bedymo will not stop with this publication. We foresee two main avenues for future development. First, we plan to include a parametrisation of moist diabatic effects. Bedymo does already contain the technical basis for such a parametrisation in the form of a passive tracer module and the model infrastructure allowing the tracer module to exert diabatic forcing on the dry dynamical core (not evaluated here). Second, we plan to complement the current Cartesian geometry in the horizontal by an option to take the spherical geometry of the Earth into account. In a longer term perspective, we envision to include an option for semi-geostrophy (Hoskins, 1975) as an intermediate step between QG and PE, as well as the option for a more dynamically active ocean component.

But already in its state at the time of publication, Bedymo represents a unique research tool. It represents an idealised complement to general circulation models run in Held-Suarez or aquaplanet simulations. Compared to these models, Bedymo omits moist diabatic effects, spherical geometry and a Hadley circulation, which is advantageous for studying all phenomena for which these aspects of the mid-latitude dynamics should not play a role. Further, due to the ease of intercomparisons between the QG and PE approximations, Bedymo is ideally suited to assess the still-unclear relation between the balanced and the unbalanced components of mid-latitude flow (cf., Plougonven and Zhang, 2014).

*Code availability.* The exact version of the model used to produce the results used in this paper is archived on Zenodo (Spensberger and Thorsteinsson, 2022), as are input data and scripts to run the model and produce the plots for all the simulations presented in this paper (Spensberger, 2022a). A user guide for the model is available as a citable pdf document (Spensberger, 2022b), and, in a more regularly updated version, at https://folk.uib.no/csp001/bedymo_doc/.

*Author contributions.* CS designed the model and prepared the manuscript, both with considerable support by TS. CS implemented the model infrastructure and atmospheric component from scratch and prepared the Figures. TT designed, implemented, and documented the slab-ocean component.

*Competing interests.* We declare no competing interests.

*Acknowledgements.* During the course of the model development, the authors have benefited strongly from discussions with a number of people. In particular we want to thank Thomas Toniazzo for in-depth mathematical and conceptual discussions on an earlier version of the model. Further, we thank Joe LaCasce, Alistair Adcroft, Michael Reeder, Robert Hallberg, Steve Garner, and Isaac Held for interesting and instructive discussion on even earlier versions of the model and QG dynamics in general. Trond Thorsteinsson was in parts supported through internal funding from the Centre for Climate Dynamics (SKD) at the Bjerknes Centre, University of Bergen.

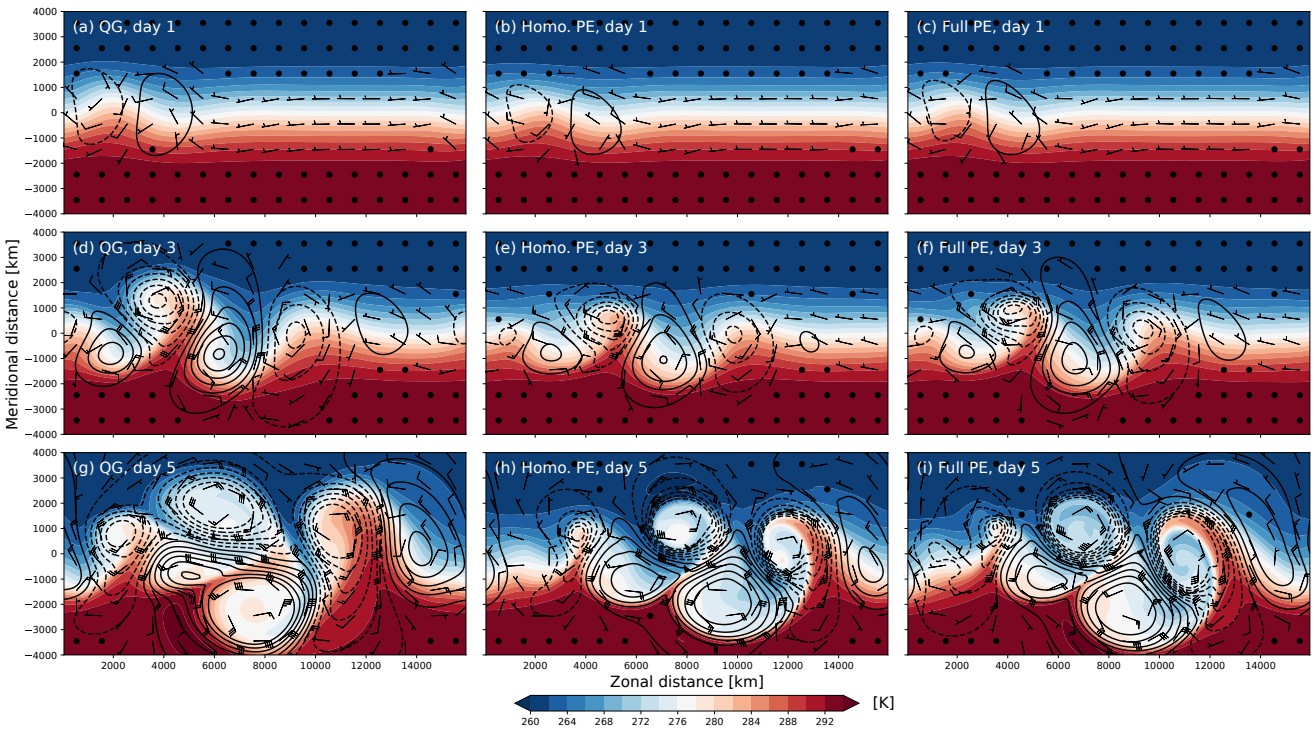

**Figure A1.** As Figure 2, but for simulations in on an $f$-plane instead of a $\beta$-plane.

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

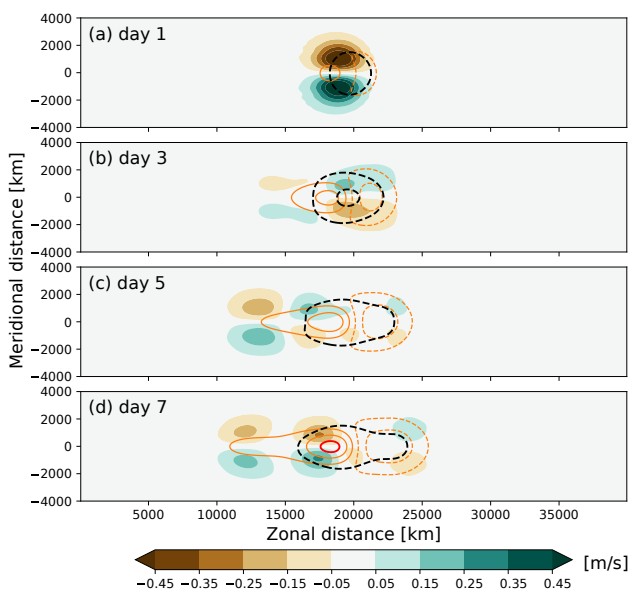

**Figure A2.** As Fig. 11, but for homogeneous density PE instead of full PE.

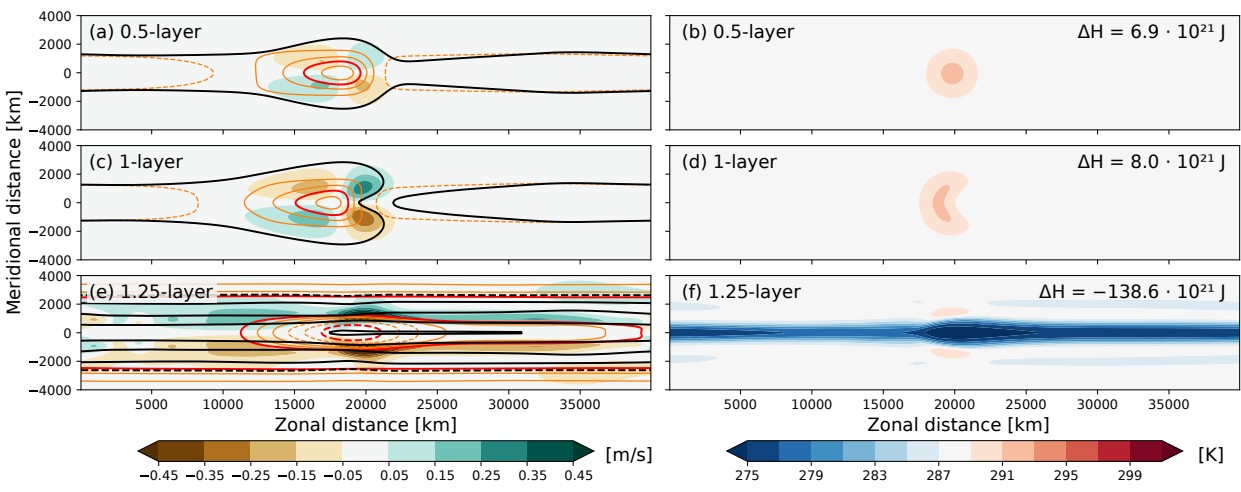

**Figure A3.** As Fig. 12, but for homogeneous density PE instead of full PE. As in Fig. 12 the maximum zonal wind anomaly on either side of the equator is about $7\,\mathrm{m\,s^{-1}}$.

Held, I. M., Ting, M., and Wang, H.: Northern Winter Stationary Waves: Theory and Modeling, Journal of Climate, 15, 2125–2144, https://doi.org/10.1175/1520-0442(2002)015<2125:NWSWTA>2.0.CO;2, 2002.

Hogg, A. M. C., Dewar, W. K., Killworth, P. D., and Blundell, J. R.: A Quasi-Geostrophic Coupled Model (Q-GCM), Monthly Weather Review, 131, 2261–2278, https://doi.org/10.1175/1520-0493(2003)131<2261:AQCMQ>2.0.CO;2, 2003.

Hoskins, B., Draghici, I., and Davies, H.: A new look at the $\omega$-equation, Quarterly Journal of the Royal Meteorological Society, 104, 31–38, 1978.

Hoskins, B. J.: The Geostrophic Momentum Approximation and the Semi-Geostrophic Equations, Journal of the Atmospheric Sciences, 32, 233–242, https://doi.org/10.1175/1520-0469(1975)032<0233:TGMAAT>2.0.CO;2, 1975.

Hoskins, B. J. and Karoly, D. J.: The Steady Linear Response of a Spherical Atmosphere to Thermal and Orographic Forcing, Journal of the Atmospheric Sciences, 38, 1179–1196, https://doi.org/10.1175/1520-0469(1981)038<1179:TSLROA>2.0.CO;2, 1981.

Matsuno, T.: Quasi-geostrophic motions in the equatorial area, J. Meteor. Soc. Japan, 44, 25–43, 1966.

Maze, G., D'Andrea, F., and de Verdière Alain, C.: Low-frequency variability in the Southern Ocean region in a simplified coupled model, Journal of Geophysical Research: Oceans, 111, n/a–n/a, https://doi.org/10.1029/2005JC003181, 2006.

McIntyre, M. E.: Spontaneous Imbalance and Hybrid Vortex–Gravity Structures, Journal of the Atmospheric Sciences, 66, 1315–1326, https://doi.org/10.1175/2008JAS2538.1, 2009.

NumPy Developers: F2PY: Fortran to Python interface generator, https://numpy.org/doc/stable/f2py/index.html, 2022.

Peixoto, J. and Oort, A. H.: Physics of Climate, 1992.

Plougonven, R. and Snyder, C.: Inertia-Gravity Waves Spontaneously Generated by Jets and Fronts. Part I: Different Baroclinic Life Cycles, Journal of the Atmospheric Sciences, 64, 2502–2520, https://doi.org/10.1175/jas3953.1, 2007.

Plougonven, R. and Zhang, F.: Internal gravity waves from atmospheric jets and fronts, Reviews of Geophysics, 52, 33–76, https://doi.org/10.1002/2012RG000419, 2014.

Rotunno, R., Muraki, D. J., and Snyder, C.: Unstable Baroclinic Waves beyond Quasigeostrophic Theory, Journal of the Atmospheric Sciences, 57, 3285–3295, https://doi.org/10.1175/1520-0469(2000)057<3285:UBWBQT>2.0.CO;2, 2000.

Saad, Y.: Iterative methods for sparse linear systems, Siam, 2003.

Schär, C. and Wernli, H.: Structure and evolution of an isolated semi-geostrophic cyclone, Quarterly Journal of the Royal Meteorological Society, 119, 57–90, 1993.

Schemm, S., Wernli, H., and Papritz, L.: Warm Conveyor Belts in Idealized Moist Baroclinic Wave Simulations, Journal of the Atmospheric Sciences, 70, 627–652, https://doi.org/10.1175/JAS-D-12-0147.1, 2013.

Simmons, A. J. and Hoskins, B. J.: The Life Cycles of Some Nonlinear Baroclinic Waves, Journal of the Atmospheric Sciences, 35, 414–432, https://doi.org/10.1175/1520-0469(1978)035<0414:TLCOSN>2.0.CO;2, 1978.

Smagorinsky, J.: On the numerical integration of the primitive equations of motion for baroclinic flow in a closed region, Monthly Weather Review, 86, 457–466, https://doi.org/10.1175/1520-0493(1958)086<0457:OTNIOT>2.0.CO;2, 1958.

Smolarkiewicz, P. K.: The Multi-Dimensional Crowley Advection Scheme, Monthly Weather Review, 110, 1968–1983, 1982.

Spensberger, C.: Bedymo configuration for simulations in model description paper, https://doi.org/10.5281/zenodo.5925424, 2022a.

Spensberger, C.: Bedymo User Guide, https://doi.org/10.5281/zenodo.5909781, 2022b.

Spensberger, C. and Thorsteinsson, T.: Bedymo: a combined quasi-geostrophic and primitive equation model in sigma coordinates, https://doi.org/10.5281/zenodo.4715686, 2022.

Terpstra, A. and Spengler, T.: An Initialization Method for Idealized Channel Simulations, Monthly Weather Review, 143, 2043–2051, https://doi.org/10.1175/mwr-d-14-00248.1, 2015.

**Table 1.** Summary of the common approach to solve the QG and PE systems. Prognostic equations are marked bold. In this table, $u$ and $v$ denote advecting wind velocities.

| Variable | QG | PE |
|---|---|---|
| $T$ | **Eq. (15)** | **Eq. (5)** |
| $p_s$ | **Eq. (16)** | **Eq. (7)** |
| $u, v$ | $= (u_g, v_g)$ | **Eqs. (1), (2)** |
| $\phi$ | Eq. (4) | Eq. (4) |
| $u_g, v_g$ | Eq. (13) | |
| $\omega, \dot{\sigma}$ | Eqs. (18), (23) | Eqs. (8), (9) |

Tierney, G., Posselt, D. J., and Booth, J. F.: An examination of extratropical cyclone response to changes in baroclinicity and temperature in an idealized environment, Climate Dynamics, 51, 3829–3846, https://doi.org/10.1007/s00382-018-4115-5, 2018.

Tremback, C. J., Powell, J., Cotton, W. R., and Pielke, R. A.: The Forward–in–Time Upstream Advection Scheme: Extension to Higher Orders, Monthly Weather Review, 115, 540–555, https://doi.org/10.1175/1520-0493(1987)115<0540:TFTUAS>2.0.CO;2, 1987.

Vallis, G. K.: Atmospheric and Oceanic Fluid Dynamics, Cambridge University Press, 2006.

Vallis, G. K., Gerber, E. P., Kushner, P. J., and Cash, B. A.: A Mechanism and Simple Dynamical Model of the North Atlantic Oscillation and Annular Modes, Journal of the Atmospheric Sciences, 61, 264 – 280, https://doi.org/10.1175/1520-0469(2004)061<0264:AMASDM>2.0.CO;2, 2004.

van der Vorst, H. A.: Bi-CGSTAB: A Fast and Smoothly Converging Variant of Bi-CG for the Solution of Nonsymmetric Linear Systems, SIAM Journal on Scientific Computing, 13, 631–644, https://doi.org/DOI:10.1137/0913035, 1992.

Voigt, A. and Shaw, T. A.: Impact of Regional Atmospheric Cloud Radiative Changes on Shifts of the Extratropical Jet Stream in Response to Global Warming, Journal of Climate, 29, 8399 – 8421, https://doi.org/10.1175/JCLI-D-16-0140.1, 2016.

Whitaker, J. S.: A Comparison of Primitive and Balance Equation Simulations of Baroclinic Waves, Journal of the Atmospheric Sciences, 50, 1519–1530, https://doi.org/10.1175/1520-0469(1993)050<1519:ACOPAB>2.0.CO;2, 1993.

Wolf, G. and Wirth, V.: Implications of the Semigeostrophic Nature of Rossby Waves for Rossby Wave Packet Detection, Monthly Weather Review, 143, 26–38, https://doi.org/10.1175/mwr-d-14-00120.1, 2015.

Table 2: List of symbols. Where ambiguous, superscripts $a$ or $o$ denote atmospheric or oceanic variables, respectively.

| Symbol | Explanation |
|---|---|
| $x, y$ | Horizontal Cartesian coordinates. |
| $\sigma$ | Vertical coordinate defined by $\sigma = \frac{p}{p_s}$. |
| $p_s$ | Surface pressure. |
| $u, v$ | Horizontal atmospheric or oceanic flow velocity components. |

| Symbol | Explanation |
| --- | --- |
| $u_g, v_g$ | Geostrophic flow velocity components. |
| $\omega, \dot{\sigma}$ | Pressure and $\sigma$ vertical velocity, respectively. |
| $u_e, v_e$ | Ekman flow velocity in the ocean. |
| $u_p, v_p$ | Prescribed flow velocity in the ocean. |
| $\boldsymbol{u}$ | Three-dimensional atmospheric or oceanic flow velocity vector. |
| $\boldsymbol{v}$ | Horizontal atmospheric or oceanic flow velocity vector. |
| $f$ | Spatially variable Coriolis parameter |
| $f_0, \beta$ | Spatially constant Coriolis and $\beta = \frac{\partial f}{\partial y}$ parameters. |
| $\overline{s}$ | Stability parameter defined in eq. (21). |
| $\alpha, \overline{\alpha}$ | Specific volume, with $\overline{\alpha} = \overline{\alpha}(\sigma)$ as with the homo. dens. approximation. |
| $\overline{T}, \overline{\Gamma}$ | Background state temperature profile and stratification. |
| $\phi$ | Hydrostatic geopotential. |
| $\psi_g$ | Geostrophic stream function defined by eq. (14). |
| $r$ | Linear (Ekman) friction coefficient. |
| $D$ | Scale-selective damping coefficient. |
| $R$ | Ideal gas constant for air. |
| $c_p, c_v$ | Isobaric and isochoric specific heat capacity for air. |
| $\phi_s$ | Time invariant surface geopotential, representing orography. |
| $J$ | Specific diabatic heating rate, unit $\mathrm{J\,kg^{-1}\,s^{-1}}$. |
| $Q_1, Q_2$ | Components of the Hoskins et al. (1978) $\boldsymbol{Q}$-vector. |
| $H$ | Depth of the slab ocean mixed layer. |
| $F_{sh}$ | Sensible heat transport between the atmosphere and ocean. |
| $\epsilon$ | Small non-rotating flow coefficient for wind-driven flow in the ocean. |

**Table 3.** Summary of pertinent model parameters for the test setups used to evaluate the model. The first four columns represent the four mid-latitude test cases discussed in sec. 3.1, with IGW being short for inertia-gravity waves. The fifth column represent the coupled test case, the Matsuno-Gill-like response to equatorial heating discussed in sec. 3.2.

| Parameter | Cyclogenesis | Storm track | Rossby waves | IGWs | Matsuno-Gill |
|---|---|---|---|---|---|
| Hor. resolution | 100 km | 100 km | 100 km | 10 km | 100 km |
| Hor. grid dimensions | 120×160 | 120×160 | 120×160 | 120×160 | 120×400 |
| Vertical levels | 3 | 3 | 10 | 30 | 3 |
| Time step QG / PE | 1200 s / 180 s | 1200 s / 180 s | 1200 s / 180 s | — / 18 s | — / 180 s |
| Simulation length | 10 days | 11 520 days | 10 days | 2 days | 60 days |
| $f_0$ | $1.117 \cdot 10^{-4}\,\mathrm{s}^{-1}$ | $1.117 \cdot 10^{-4}\,\mathrm{s}^{-1}$ | $1.117 \cdot 10^{-4}\,\mathrm{s}^{-1}$ | $1.117 \cdot 10^{-4}\,\mathrm{s}^{-1}$ | 0 |
| $\beta$ | $1.472 \cdot 10^{-11}\,(\mathrm{s\,m})^{-1}$ | $1.472 \cdot 10^{-11}\,(\mathrm{s\,m})^{-1}$ | $1.472 \cdot 10^{-11}\,(\mathrm{s\,m})^{-1}$ | 0 | $1.472 \cdot 10^{-11}\,(\mathrm{s\,m})^{-1}$ |
| $r$ | 0 | $1.0 \cdot 10^{-6}\,\mathrm{s}^{-1}$ | 0 | 0 | 0 |
| $D$ | 0 | $1.0 \cdot 10^{-5}\,\mathrm{m}^2\,\mathrm{s}^{-1}$ | $1.0 \cdot 10^{-4}\,\mathrm{m}^2\,\mathrm{s}^{-1}$ | $1.0 \cdot 10^{-4}\,\mathrm{m}^2\,\mathrm{s}^{-1}$ | $1.0 \cdot 10^{-5}\,\mathrm{m}^2\,\mathrm{s}^{-1}$ |
| Temp. relax. coeff. | 0 | $1.0 \cdot 10^{-6}\,\mathrm{s}^{-1}$ | 0 | 0 | 0 |

| Symbol | Explanation |
|---|---|
| $C_{sh}, C_D$ | Exchange coefficients for sensible heat and momentum fluxes. |