# Peer review of "Bedymo: a combined quasi-geostrophic and primitive equation model in sigma coordinates"

_Geoscientific Model Development, 2021_

## Referee Comment (RC2)

Response on "Bedymo: a combined quasi-geostrophic and primitive equation" by Clemens Spensberger, Trond Thorsteinsson, and Thomas Spengler

The study introduces a new modeling framework, aimed at solving the QG and PE equations as closely as possible. The QG model does not capture the dynamics outside the midlatitudes accurately and the primitive equation solvers are mostly formulated in the context of simulating the global circulation and are thus constructed differently making a direct comparison with a QG model elusive. The framework, designed in cartesian coordinates, is tested using 4 tests including both the atmosphere and the coupled atmosphere-ocean. The tests are carefully designed and succeed in demonstrating the strong similarities and differences between QG and PE by analyzing responses in baroclinic growth, coupling and stationary multi-wave forcings. I detail some fairly minor and some major comments below that can help improve the manuscript. I suggest accepting the manuscript, pending these revisions.

**1 Minor Comments**

1. Using sigma-coordinates in the vertical is common practice among models. Please either mention why is using sigma-cordinate in the vertical unique? or remove this sentence altogether.

2. L15 : "1950s"

3. L42 : as well → separately

4. L244 : should be "in line with"

5. In Figures 1,3,6,7 and 8, the x and y labels are interchanged. The x axis is the zonal distance and the y axis is the meridional distance. This should be corrected.

6. L246-7 : ... "results of Simmons and Hosking (1978), for instance, which" ...

7. Section 3.1.2 : maybe I am missing something but how many levels are you using for the storm track test case? Were the original 26 vertical levels used for the PE case?

8. : I am confused by the writing here. There has to be zonal symmetries because you are computing the eddy covariances next. Please reconcile this. The stork tracks are not symmetric - so please revise the sentence.

9. L276 : and the ones actually observed.

10. L296 : , however, do not

**2 Major Comments**

1. I suggest a re-writing of the abstract because it is misleading with regards to the contents of the manuscript. There is no mention of the tracers in the manuscript, only the abstract. Thus, the sentence should be removed. There is no discussion about the graphical interface in the manuscript either, just a brief mention in the introduction section.

2. The Introduction is inadequately written and I suggest revising it. No historical or existing literature has been discussed. It's severely lacking any conviction on the importance of the study and how it fits within recent modeling initiatives, apart from the mention that the models are being solved as closely as possible.

(a) Please discusses key differences between QG and PE and the traditional methods used to numerically solve them.

(b) Since this is the key foundation the manuscript stands on, please also elaborate the key differences between a QG model and the PE model, when used for atmospheric analysis (apart from the well-known fact that the QG framework works well only in the midlatitudes).

(c) Horizontal and vertical coupling in models is challenging. Moreover, diffusion, which affects momentum balance strongly on long timescales, can also affect the model performance and is strongly dependent on the employed numerical schemes. In a QG model using sigma coordinates, it can introduce strong ageostrophic fluxes. Acknowledging these issues is important in the introduction and so is connecting to the other well established studies aimed at testing dynamical cores, to put the study into a proper perspective. I recommend adding some discussion connecting this study to the more recent studies on evaluation of dynamical cores in much more detail :

Lin, S.-J., Harris, L., Chen, X., Yao, W. & Chai, J. (2017). Colliding modons: A nonlinear test for the evaluation of global dynamical cores. Journal of Advances in Modeling Earth Systems, 9, 2483 2492. https://doi.org/10.1002/2017MS000965

Gupta, A, Gerber, EP, Lauritzen, PH. Numerical impacts on tracer transport: A proposed intercomparison test of Atmospheric General Circulation Models. Q J R Meteorol Soc. 2020; 146: 3937 3964. https://doi.org/10.1002/qj.3881

Ma, J., Xu, S., & Wang, B. (2020). Reducing numerical diffusion in dynamical coupling between atmosphere and ocean in Community Earth System Model version 1.2.1. Journal of Advances in Modeling Earth Systems, 12, e2020MS002052. https://doi.org/10.1029/2020MS002052

Held, I. M., & Suarez, M. J. (1994). A Proposal for the Intercomparison of the Dynamical Cores of Atmospheric General Circulation Models, Bulletin of the American Meteorological Society, 75(10), 1825-1830.

3. Since the model is developed ground up and prepared for educational and research purposes, it is important to provide a performance analysis with a traditional dynamical core. Thus, I suggest the authors to add another a discussion/figure and/or qualitatively compare the runtime of Bedymo with a Held-Suarez like model, for instance, with similar numerical schemes.

4. The study would make for a much more valuable contribution to the existing literature if there was a functionality to use the PE framework in spherical coordinates as well, considering the overall curvature of the planet. Is that a possibility with the framework so far? If not, then the use of the model might be quite limited.

5. I strongly encourage the authors to elaborate more on the future possibilities using the model in the conclusion/discussion section. Is there a plan to introduce more features in the model (this might be a good point to talk briefly about the tracer module)? How does having a "live" graphical interface in the model give it an edge over the traditional frameworks (even modern dynamical cores from prominent modeling centered allow visualisation using certain software packages)?

---

## Author Comment (AC1)

**Response to reviews**

We thank the two anonymous referees and Sebastian Fromang for their constructive feedback on the manuscript and the model it describes. We are happy to see that the overall approach was generally well-received, and will make sure to address the issues and fill the gaps that were pointed out in the revised version of the manuscript. Point-by-point responses to the specific comments are below, with our response appearing in blue.

**Comments from Sebastien Fromang**

The paper presents the code Bedymo, which aims at implementing as similarly as possible two approximations of the fluid equations that are widely used in atmospheric sciences, namely the quasi-geostrophic and the primitive equations. Although there exists many implementations of these equations in many different codes developped by many different teams, the main interest of Bedymo is to try to make these two implementations as similarly as possible in one single framework, so as to be able to compare as closely as possible the role of the different terms of the equations in shaping the atmospheric flow. The paper is divided in two parts: the first part (section 2) presents the two sets of equations. Although I found this part a bit arid and sometimes difficult to follow, I understand it is due to the nature of that section and the need to be complete, and I have to admit I have no suggestion to make it better...The second part of the paper (section 3) presents a series of standard tests that evaluate and validate the implementation.

The results are interesting and worth publishing in GMD. I have some comments I detail below, but they do not require significant modifications of the paper. The only thing I feld was missing when I read the paper was some sort of "hands-on" section (see also my point 2 below). Such a section could be very useful for students or unexperienced users that would want to try using Bedymo. It could be thought of as some sort of small tutorial that would give the minimal system requirements and illustrate the typical few steps one would have to follow to run a first simple simulation (downloading, compiling, running code and visualizing the results). It would also be a good opportunity to describe and illustrate the graphical user interface that is mentioned in the abstract.

To summarize, I will be happy to recommend publication in GMD after the minor revisions I described below are taken care of.

1.  In the abstract, the authors mention the existence of a "slab ocean model and passive tracer module that will provide the basis for future idealised parametrisation of moisture and latent heat release". Unless I am wrong and missed it, the passive tracer module is never mentioned and is not validated in the paper. I would suggest to remove from the abstract that part of the sentence. It could be briefly mentioned in the paper conclusions though (see my point 8 below).

    We agree and will remove the tracer module from the abstract. Based on comments from several reviewers we will introduce a new outlook section and will make sure to mention the existence of a tracer module there.

2.  Likewise, in the abstract, the authors state: "Bedymo has a graphical user interface, making it particularly useful for teaching". The graphical user interface is also mentioned in one sentence at the end of the introduction, where it is said also that the "python bindings (...) provide the basis to watch the flow evolution live while the

model is running". This seems like an interesting feature. But again, it is not illustrated in the main body of the paper, which is unfortunate.

We agree and will include a screenshot of the GUI in the main manuscript.

This suggests that the author might want to add some sort of additional tutorial section (as already mentioned above), where they could give some sort of cookbook for a student that would want to use the model on a linux platform: how and where should I download and compile the code? how can I setup, in practice, the simplest simulation? and, next, how should I setup and use the GUI interface? This would also be a good opportunity to show a few snapshots to illustrate the GUI interface...I feel such an hands-on practical section is missing for a paper describing a code that is claimed to be user-friendly and easy to use for teaching!

We agree here, too, but feel that including a user guide would be beyond the scope of this article. Nevertheless, the reviewer is clearly right in that we need such a user guide to simplify the use of the model outside Bergen. We will thus make available a user guide through a website and a citable pdf document separately along with (and cited from) the revised manuscript.

3. I find the caption of figure 3 unclear. According to the text (line 235-236), I was expecting to see snapshots of different simulations, with different jet speeds. This is the case, isn't it? Yet, if one only reads the caption, it only says that the figure shows the "sensitivity of the downstream development....", but does not say to what....I think the caption should be rewritten, and explicitely says which initial jet speed is used for the different panels. Likewise, it would be useful to mention in the caption of figure 1 the jet speed used for the simulations shown in that figure.

Apologies for the confusion, we will clarify the caption. The reviewer has interpreted the Figure correctly, though, the Figure shows the sensitivity of downstream development to the initial jet speed.

4. Commenting the result of the rossby wave excited by orography, the authors end section 3.1.3 (line 313) saying that "the response fits qualitatively well to the response expected from linear wave theory" and cite, e.g. the paper by Hoskins & Karoly. Could the authors be a bit more explicit in describing those aspects of the flow that agree with the theory? Likewise, would it be possible to go a bit further and make this comparison more quantitative? For example, by comparing the wavelength of the Rossby wave expected from the theory with that obtained in the simulation? and/or by comparing the Rossby wave path expected from the theory with that obtained in the simulation?

We agree with the reviewer that this comparison appears somewhat superficial. We will in the revised manuscript include a new Figure (or panel) showing the linear stationary QG solution as a reference, thus allowing direct and quantitative comparsion of the solutions.

5. Line 351-352: The authors state: "Overall this leads to a slightly larger ocean heat content with the 1.0 layer compared to the 0.5-layer ocean (compare fig.8d and fig.8b)": I find that the comparison between the two mentioned panels of figure 8 is not enlightning....Maybe this is because the difference between the ocean heat

content is only "slightly" different, or because the contour levels in figure 8 are not well chosen, but I find this difference is not apparent when I compare fig 8b and 8d. I see the shape of the SST anomaly is different. It is not so clear that the heat content associated with these anomalies is different....Maybe it would be better to compute that difference numerically and give it in the text? It could either be the absolute difference, or the relative difference between the two cases....
We agree, and will include this information as a number in each Figure panel.

6. There is a problem in the second line of Fig. 8 caption. It says: "the rows show the development of (a) the 0.5 layer model, (b) the 1-layer model, and (c) the 1.25 layer model". It should be: "the rows show the development of (a-b) the 0.5 layer model, (c-d) the 1-layer model, and (e-f) the 1.25 layer model"
Thanks for pointing out this mistake, we will correct.

7. For the coupled test in general, and particularly for the 1 and 1.25 layer model, I am a bit puzzled because there is not comparison with any expected results or anything from the literature (as opposed to the 0.5 layer model case). This raises the question of whether these tests are useful at all in the present paper? Except for showing that these two versions of the slab ocean model are running w/o crashing, I don't see to what extent they validate the implementation of these versions of the slab model. Could the authors clarify that point? If the results cannot be validated, I am tempted to suggest to remove them, given the focus of the paper which is to validate Bedymo implementation.
The reviewer is right, the variants of the test case with interative ocean go beyond what could be captured in and thus evaluated against linear models. We nevertheless included them to showcase the additional dynamics introduced by several simple variants of a slab-ocean model. Due to the limitation of the linear model, we are here limited to plausibility checks—which we think the slab ocean model passes well. We will make sure to point out this caveat in the evaluation explicitly in the manuscript.

8. Finally, I would suggest to add a paragraph in section 4 that would be focused on the perspectives of the code: what will be the main use of Bedymo (in its current version) in the next few years? What are the perspectives for its evolution? There, for example, the tracer module and its possible use to implement simple parametrization in the future could be mentioned.
We fully agree with this and similar comments by the other reviewers and will include such an outlook section.

**Comments from Anonymous Referee #1**

The quasi-geostrophic (QG) model has been developed to study the large-scale flow and is one of the most successful models in meteorology. It did not only allow for the first successful numerical weather forecasts but it can be used to explain almost all features we find in large-scale flows (see e.g. Pedlosky 1987). However, more complete models have been developed to describe phenomena beyond the QG scaling, e.g. primitive equation (PE) models. It is important and was done by several earlier studies, to compare the QG and PE

dynamics to validate models and to understand better processes consistent or beyond QG dynamics.

The manuscript entitled 'Bedymo: a combined quasi-geostrophic and primitive equation model in sigma coordinates' by Clemens Spensberger, Trond Thorsteinsson, and Thomas Spengler is about the combination of the quasi-geostrophic approximation and the hydrostatic primitive equations in one modelling framework. Different case studies are done: baroclinic life cycles, storm tracks, topographic flow, and equatorial waves. The cases are described well and show the power and quality of the models. I have a number of comments the authors might consider before publication.

Comments:

1. In principle, at least over a certain time period, each model can be run in a "QG mode" by just respecting the QG assumptions. The Rossby number needs to be small, the topography needs to be shallow, etc. Of course, a PE model will develop spatio-temporal ageostrophic dynamics when small scales are not filtered. I wonder, whether this resolution aspect has been considered. When comparing QG and PE, was the size of the time step and the spatial resolution the same? I think this issue is of particular importance for the storm track simulations.

   The domain setup and resolution of the QG and PE simulation is identical for each test case, respectively, in order to make the two solutions as comparable as possible. Only the time step is different for reasons that will become clear in the following. Both the QG and the PE systems include ageostrophic flow; the difference is that in QG the ageostrophic flow can be diagnostically deduced from the balanced geostrophic flow, whereas in PE the ageostrophic flow is indirectly implied in the tendency equations of horizontal momentum and temperature. Consequently, the PE ageostrophic flow is much less clearly tied to the balanced geostrophic flow than in QG and contains, for example, baroclinic and barotropic inertia gravity waves (IGWs). We here want to stress that the difference between QG and PE ageostrophic flow is not primarily a matter of scale; for example, some of the IGWs have wave lengths and amplitudes that might seem consistent with the assumptions of QG, yet a QG model can still not represent them. Barotropic IGWs propagate very fast (> 300 m/s) and we thus need to adapt the time step in the PE model to accommodate them.

2. The values of the chosen parameters should be given (time step, resolution,...). Also all the values for the coefficients in the models should be given: r, D, alpha,.... In particular, some of the coefficients occur in the QG and the PE model (e.g. r and D). Are they the same in both models?

   The parameters are consistent between QG and PE as far as possible. In particular, resolution and the physical coefficients are the same and only the time step differs to accommodate the fast propagation of IGWs in PE (cf. response to comment 1). We will introduce a new table summarising all parameters and coefficients.

3. I think Bedymo could be very helpful to study the concept of balance. Recently, interest has increased significantly to better understand the coupling of the slow and the fast dynamics. QG solutions are balanced solutions without any internal gravity waves. Implementing such solutions into a PE model will destroy the balance. Bedymo would be an ideal tool for studying such processes, e.g. so called

spontaneous imbalance. This means, however, to resolve the PE model properly. An overview on this issue can be found in a JAS special collection at https://journals.ametsoc.org/collection/spontaneous- imbalance .

We fully agree with this remark and thank the reviewer for bringing up this literature as context for our work. We will make sure to include it in relevant sections of the manuscript, first and foremost in the introduction and a new outlook section based on different comments from all reviewers.

4. The cases considered test mainly the quality of the QG model and the ability of the PE model to represent the QG solutions. Has the PE model also been tested against typical PE test cases?

We thank the reviewer for pointing out this slight gap in our evaluation. We will in the revised manuscript include a new test case focussing on inertia-gravitiy waves excited by isolated orography in a mid-latitude channel.

5. The color code of Fig. 2, 4, 5 is not easy to read. Using different line styles might make the figures more clear.

We agree and will adapt the Figures to make them more easily readable.

6. I think for the baroclinic life cycle and the storm track case the boundaries in the meridional direction are closed (v=0). However, for the Rossby wave case these boundaries seem to be open. Could the authors give a short description how the open radiative boundaries have been implemented in both models?

We unfortunately do not yet have the option for open boundaries in Bedymo. The model domain extends beyond the shown plot domain, as noted in the Figure caption.

7. What is the reason for the asymmetry in the PE Rossby wave case?

The slight asymmetries are introduced by IGWs excited by the shock of the initial flow adjusting to the topography. We will experiment with increased damping to remove these waves over time, and/or mention the reason for the asymmetries in the manuscript.

8. For the coupled case a equatorial flow has been chosen that cannot be compared to QG dynamics that covers mid-latitude flows only. Instead the QG comparison, the PE solutions are compared with analytical solutions from linear wave theory. This works only for sufficiently small heating. Was the local heating chosen comparable to the sources used e.g. in the paper by Gill (given in the reference)?

Thanks for raising this point. The initial SST anomalies peaks at +5K, thus introducing a clearly finite-amplitude heating in the atmosphere. We will mention this caveat in the discussion of this test case.

For the linear theories, the ocean was passive. Was the motivation to couple an active surface layer ocean to study the differences?

The reviewer is right, the variants of the test case with interative ocean go beyond what could be captured in and thus evaluated against linear models. We nevertheless included them to showcase the additional dynamics introduced by

several simple variants of a slab-ocean model. With the limitations of the linear model, we are here limited to plausibility checks—which we think the slab ocean model passes well. We will make sure to point out this caveat in the evaluation explicitly in the manuscript.

9. Recently, very interesting nonlinear solutions of equatorial waves have been documented (see e.g. Rostami, M., and Zeitlin, V. "Eastward-moving equatorial modons: a missing chain-link in the dynamics of the tropical atmosphere?", Phys. Fluids,31, 021701, 2019). For future work it would be very instructive to try to find such solutions with the PE model.
We agree, and will keep this reference in mind for potential extensions of this work.

**Comments from Anonymous Referee #2**

The study introduces a new modeling framework, aimed at solving the QG and PE equations as closely as possible. The QG model does not capture the dynamics outside the midlatitudes accurately and the primitive equation solvers are mostly formulated in the context of simulating the global circulation and are thus constructed differently making a direct comparison with a QG model elusive. The framework, designed in cartesian coordinates, is tested using 4 tests including both the atmosphere and the coupled atmosphere-ocean. The tests are carefully designed and succeed in demonstrating the strong similarities and differences between QG and PE by analyzing responses in baroclinic growth, coupling and stationary multi-wave forcings. I detail some fairly minor and some major comments below that can help improve the manuscript. I suggest accepting the manuscript, pending these revisions.

**Minor Comments**

1. Using sigma-coordinates in the vertical is common practice among models. Please either mention why is using sigma-cordinate in the vertical unique? or remove this sentence altogether.
   The use of sigma-coordinates is certainly common among PE models, but is unique among QG models, which are most often formulted in pressure or log(pressure)-coordinates. We will clarify the statement.

2. L15 : "1950s"
3. L42 : as well → separately
4. L244 : should be "in line with"
5. In Figures 1,3,6,7 and 8, the x and y labels are interchanged. The x axis is the zonal distance and the y axis is the meridional distance. This should be corrected.
6. L246-7 : ... "results of Simmons and Hosking (1978), for instance, which" ...
   Thanks for pointing out the above mistakes, we will correct.

7. Section 3.1.2 : maybe I am missing something but how many levels are you using for the storm track test case? Were the original 26 vertical levels used for the PE case?
   Thanks for pointing out this missing information. We will add it to the manuscript as well as a new table summarising model parameters. Unless noted otherwise (e.g. for the bartropic Rossby wave test case), the model was run with 3 vertical levels. This is

in line with other typical QG setups (cf. for example Fromang and Riviere 2020, doi: 10.1175/JAS-D-19-0178.1).

8. I am confused by the writing here. There has to be zonal symmetries because you are computing the eddy covariances next. Please reconcile this. The stork tracks are not symmetric - so please revise the sentence.
Unfortunately the line number to which this comment corresponds did not make it through in the review document. We assume you refer to L261, "Because of the absence of any zonal asymmetries, the resulting storm track is zonally symmetric." As far as we can see, this might be a misunderstanding as we here only talk about the "storm track", i.e. the eddy statistics aggregated over time, being zonally symmetric. This does not imply that, for example, the instantaneous wind velocity components are zonally symmetric as well, they are certainly not. We will make this clearer in the revised version of our manuscript.

9. L276 : and the ones actually observed.
10. L296 : , however, do not
Thanks again for pointing out these mistakes, we will correct them.

**Major Comments**

1. I suggest a re-writing of the abstract because it is misleading with regards to the contents of the manuscript. There is no mention of the tracers in the manuscript, only the abstract. Thus, the sentence should be removed. There is no discussion about the graphical interface in the manuscript either, just a brief mention in the introduction section.
Thanks for pointing out this mismatch between abstract and manuscript. To resolve this, we will remove the tracer module from the abstract and include a screenshot of the GUI in the manuscript.

2. The Introduction is inadequately written and I suggest revising it. No historical or existing literature has been discussed. It's severely lacking any conviction on the importance of the study and how it fits within recent modeling initiatives, apart from the mention that the models are being solved as closely as possible.
(a) Please discusses key differences between QG and PE and the traditional methods used to numerically solve them.
(b) Since this is the key foundation the manuscript stands on, please also elaborate the key differences between a QG model and the PE model, when used for atmospheric analysis (apart from the well-known fact that the QG framework works well only in the midlatitudes).
Thanks for pointing out these missing aspects in the introduction. We will make sure to include them in the revised manuscript.

(c) Horizontal and vertical coupling in models is challenging. Moreover, diffusion, which affects momentum balance strongly on long timescales, can also affect the model performance and is strongly dependent on the employed numerical schemes. In a QG model using sigma coordinates, it can introduce strong ageostrophic fluxes. Acknowledging these issues is important in the introduction and so is connecting to

the other well established studies aimed at testing dynamical cores, to put the study into a proper perspective.

We agree with the reviewer in that numerical diffusion can be an issue that degrades model simulations. We have implemented in Bedymo advection schemes of 1st to 6th order, with the odd-order schemes being upwind biased. We have systematically tested the implementation of these schemes, for example using rotating a cone as a test case. In these tests, all implementations worked as expected, but of course lower-order schemes turned out considerably more diffusive than higher-order schemes. For our physical test cases, the choice of the advection scheme however turned out largely inconsequential as the explicit diffusion is dominating over the numerical diffusion. We thank the reviewer for pointing out this gap in the description of the model numerics and we will make sure to discuss this point in the revised manuscript.

I recommend adding some discussion connecting this study to the more recent studies on evaluation of dynamical cores in much more detail :

- Lin, S.-J., Harris, L., Chen, X., Yao, W. & Chai, J. (2017). Colliding modons: A nonlinear test for the evaluation of global dynamical cores. Journal of Advances in Modeling Earth Systems, 9, 2483 2492. https://doi.org/10.1002/2017MS000965
- Gupta, A, Gerber, EP, Lauritzen, PH. Numerical impacts on tracer transport: A pro- posed intercomparison test of Atmospheric General Circulation Models. Q J R Meteorol Soc. 2020; 146: 3937 3964. https://doi.org/10.1002/qj.3881
- Ma, J., Xu, S., & Wang, B. (2020). Reducing numerical diffusion in dynamical coupling between atmosphere and ocean in Community Earth System Model version 1.2.1. Journal of Advances in Modeling Earth Systems, 12, e2020MS002052. https://doi.org/10.1029/2020MS002052
- Held, I. M., & Suarez, M. J. (1994). A Proposal for the Intercomparison of the Dynamical Cores of Atmospheric General Circulation Models, Bulletin of the American Meteorological Society, 75(10), 1825-1830.

We thank the reviewer for these pointers. We will include and discuss these references as far as and as detailed as relevant for our manuscript. The discussion might be relatively brief, because, as argued above, numerical diffusion is not a primary concern for Bedymo. Further, as the reviewer noted, we leave Bedymo's tracer module unevaluated in this manuscript and will in the revised version thus only mention it in a new outlook section.

3. Since the model is developed ground up and prepared for educational and research purposes, it is important to provide a performance analysis with a traditional dynamical core. Thus, I suggest the authors to add another a discussion/figure and/or qualitatively compare the runtime of Bedymo with a Held-Suarez like model, for instance, with similar numerical schemes.

We would have very much liked to do such a comparision, but contrary to intuition there are no other established dynamical cores that are similar enough to make such a comparision insightful. Held-Suarez-type models are global and such include a Hadley circulation and thus subtropical jets, whereas the jets in Bedymo by construction can only be eddy-driven. Further, Held-Suarez-type models need to

take into account the spherical geometry of the Earth, which Bedymo does not. For these reason, a comparision of, for example the wind speed in Figure 2 of Held and Suarez (1994) with our Figure 4 cannot really evaluate whether Bedymo is doing the right thing for the right reason. Nevertheless, we agree with the reviewer that the current discussion of Held-Suarez is too superficial, and we will expand and mention the above caveats with the comparison.

4. The study would make for a much more valuable contribution to the existing literature if there was a functionality to use the PE framework in spherical coordinates as well, considering the overall curvature of the planet. Is that a possibility with the framework so far? If not, then the use of the model might be quite limited.

   For now, Bedymo does not support a spherical geometry. An extension towards spherical geometry is one of our plans for further development, and we will mention it in the new outlook section.

5. I strongly encourage the authors to elaborate more on the future possibilities using the model in the conclusion/discussion section. Is there a plan to introduce more features in the model (this might be a good point to talk briefly about the tracer module)? How does having a "live" graphical interface in the model give it an edge over the traditional frameworks (even modern dynamical cores from prominent modeling centered allow visualisation using certain software packages)?

   We agree with the reviewer, and include these points
   a. by including and discussing a screenshot of the graphical user interface in the manuscript,
   b. by a new outlook section detailling plans and perspectives for future development, and
   c. by providing and citing a separate user guide that describes how to setup and run the model.

---

## Author Response (AR1)

**Response to reviews**

We thank the two anonymous referees and Sebastian Fromang for their constructive feedback on the manuscript and the model it describes. We are happy to see that the overall approach was generally well-received, and will make sure to address the issues and fill the gaps that were pointed out in the revised version of the manuscript. Point-by-point responses to the specific comments are below, with our response appearing in blue. Line numbers in our response refer to the difference document between the original and the revised submission.

**Comments from Sebastien Fromang**

The paper presents the code Bedymo, which aims at implementing as similarly as possible two approximations of the fluid equations that are widely used in atmospheric sciences, namely the quasi-geostrophic and the primitive equations. Although there exists many implementations of these equations in many different codes developped by many different teams, the main interest of Bedymo is to try to make these two implementations as similarly as possible in one single framework, so as to be able to compare as closely as possible the role of the different terms of the equations in shaping the atmospheric flow. The paper is divided in two parts: the first part (section 2) presents the two sets of equations. Although I found this part a bit arid and sometimes difficult to follow, I understand it is due to the nature of that section and the need to be complete, and I have to admit I have no suggestion to make it better...The second part of the paper (section 3) presents a series of standard tests that evaluate and validate the implementation.

The results are interesting and worth publishing in GMD. I have some comments I detail below, but they do not require significant modifications of the paper. The only thing I feld was missing when I read the paper was some sort of "hands-on" section (see also my point 2 below). Such a section could be very useful for students or unexperienced users that would want to try using Bedymo. It could be thought of as some sort of small tutorial that would give the minimal system requirements and illustrate the typical few steps one would have to follow to run a first simple simulation (downloading, compiling, running code and visualizing the results). It would also be a good opportunity to describe and illustrate the graphical user interface that is mentioned in the abstract.

To summarize, I will be happy to recommend publication in GMD after the minor revisions I described below are taken care of.

1. In the abstract, the authors mention the existence of a "slab ocean model and passive tracer module that will provide the basis for future idealised parametrisation of moisture and latent heat release". Unless I am wrong and missed it, the passive tracer module is never mentioned and is not validated in the paper. I would suggest to remove from the abstract that part of the sentence. It could be briefly mentioned in the paper conclusions though (see my point 8 below).

   We agree and removed the tracer module from the abstract. Based on comments from several reviewers we introduced a new outlook section (sec. 5) and mention the existence of a tracer module there.

2. Likewise, in the abstract, the authors state: "Bedymo has a graphical user interface, making it particularly useful for teaching". The graphical user interface is also

mentioned in one sentence at the end of the introduction, where it is said also that the "python bindings (...) provide the basis to watch the flow evolution live while the model is running". This seems like an interesting feature. But again, it is not illustrated in the main body of the paper, which is unfortunate.

We agree and included a screenshot of the GUI in the main manuscript (new Fig. 1).

This suggests that the author might want to add some sort of additional tutorial section (as already mentioned above), where they could give some sort of cookbook for a student that would want to use the model on a linux platform: how and where should I download and compile the code? how can I setup, in practice, the simplest simulation? and, next, how should I setup and use the GUI interface? This would also be a good opportunity to show a few snapshots to illustrate the GUI interface...I feel such an hands-on practical section is missing for a paper describing a code that is claimed to be user-friendly and easy to use for teaching!

We agree here, too, but feel that including a user guide would be beyond the scope of this article. Nevertheless, the reviewer is clearly right in that we need such a user guide to simplify the use of the model outside Bergen. We thus made available a user guide through a website (https://folk.uib.no/csp001/bedymo_doc/) and a citable pdf document (DOI: 10.5281/zenodo.5909781) which is cited from the revised manuscript.

3. I find the caption of figure 3 unclear. According to the text (line 235-236), I was expecting to see snapshots of different simulations, with different jet speeds. This is the case, isn't it? Yet, if one only reads the caption, it only says that the figure shows the "sensitivity of the downstream development....", but does not say to what....I think the caption should be rewritten, and explicitly says which initial jet speed is used for the different panels. Likewise, it would be useful to mention in the caption of figure 1 the jet speed used for the simulations shown in that figure.

Apologies for the confusion, we rewrote the caption and added the missing information. The reviewer interpreted the Figure correctly, the Figure shows the sensitivity of downstream development to the initial jet speed.

4. Commenting the result of the rossby wave excited by orography, the authors end section 3.1.3 (line 313) saying that "the response fits qualitatively well to the response expected from linear wave theory" and cite, e.g. the paper by Hoskins & Karoly. Could the authors be a bit more explicit in describing those aspects of the flow that agree with the theory? Likewise, would it be possible to go a bit further and make this comparison more quantitative? For example, by comparing the wavelength of the Rossby wave expected from the theory with that obtained in the simulation? and/or by comparing the Rossby wave path expected from the theory with that obtained in the simulation?

We agree with the reviewer that this comparison appears somewhat superficial. We included a new Figure 8 in the revised manuscript showing the analytic linear stationary QG solution as a reference, thus allowing direct and quantitative comparison of the solutions. The results from Bedymo generally agree well with the analytic solution.

5.  Line 351-352: The authors state: "Overall this leads to a slightly larger ocean heat content with the 1.0 layer compared to the 0.5-layer ocean (compare fig.8d and fig.8b)": I find that the comparison between the two mentioned panels of figure 8 is not enlightning....Maybe this is because the difference between the ocean heat content is only "slightly" different, or because the contour levels in figure 8 are not well chosen, but I find this difference is not apparent when I compare fig 8b and 8d. I see the shape of the SST anomaly is different. It is not so clear that the heat content associated with these anomalies is different....Maybe it would be better to compute that difference numerically and give it in the text? It could either be the absolute difference, or the relative difference between the two cases....
    We agree and include the total mixed layer heat anomaly relative to 288 K as a number in each panel of Figs. 11 and A3 in the revised manuscript.

6.  There is a problem in the second line of Fig. 8 caption. It says: "the rows show the development of (a) the 0.5 layer model, (b) the 1-layer model, and (c) the 1.25 layer model". It should be: "the rows show the development of (a-b) the 0.5 layer model, (c-d) the 1-layer model, and (e-f) the 1.25 layer model"
    Thanks for pointing out this mistake, we corrected it.

7.  For the coupled test in general, and particularly for the 1 and 1.25 layer model, I am a bit puzzled because there is not comparison with any expected results or anything from the literature (as opposed to the 0.5 layer model case). This raises the question of whether these tests are useful at all in the present paper? Except for showing that these two versions of the slab ocean model are running w/o crashing, I don't see to what extent they validate the implementation of these versions of the slab model. Could the authors clarify that point? If the results cannot be validated, I am tempted to suggest to remove them, given the focus of the paper which is to validate Bedymo implementation.
    The reviewer is right, the variants of the test case with interactive ocean go beyond what could be captured in and thus evaluated against linear models. We nevertheless included them to showcase the additional dynamics introduced by several simple variants of a slab-ocean model. Due to the limitation of the linear model, we are here limited to plausibility checks—which we think the slab ocean model passes well. We now point out this caveat in the evaluation explicitly in the manuscript (L431-433).

8.  Finally, I would suggest to add a paragraph in section 4 that would be focused on the perspectives of the code: what will be the main use of Bedymo (in its current version) in the next few years? What are the perspectives for its evolution? There, for example, the tracer module and its possible use to implement simple parametrization in the future could be mentioned.
    We fully agree with this and similar comments by the other reviewers and now include an outlook section to this end (new sec. 5).

**Comments from Anonymous Referee #1**

The quasi-geostrophic (QG) model has been developed to study the large-scale flow and is one of the most successful models in meteorology. It did not only allow for the first successful numerical weather forecasts but it can be used to explain almost all features we find in large-scale flows (see e.g. Pedlosky 1987). However, more complete models have been developed to describe phenomena beyond the QG scaling, e.g. primitive equation (PE) models. It is important and was done by several earlier studies, to compare the QG and PE dynamics to validate models and to understand better processes consistent or beyond QG dynamics.

The manuscript entitled 'Bedymo: a combined quasi-geostrophic and primitive equation model in sigma coordinates' by Clemens Spensberger, Trond Thorsteinsson, and Thomas Spengler is about the combination of the quasi-geostrophic approximation and the hydrostatic primitive equations in one modelling framework. Different case studies are done: baroclinic life cycles, storm tracks, topographic flow, and equatorial waves. The cases are described well and show the power and quality of the models. I have a number of comments the authors might consider before publication.

Comments:

1. In principle, at least over a certain time period, each model can be run in a "QG mode" by just respecting the QG assumptions. The Rossby number needs to be small, the topography needs to be shallow, etc. Of course, a PE model will develop spatio-temporal ageostrophic dynamics when small scales are not filtered. I wonder, whether this resolution aspect has been considered. When comparing QG and PE, was the size of the time step and the spatial resolution the same? I think this issue is of particular importance for the storm track simulations.

   The domain setup and resolution of the QG and PE simulation is identical for each test case, respectively, in order to make the two solutions as comparable as possible. Only the time step is different for reasons that will become clear in the following. Both the QG and the PE systems include ageostrophic flow; the difference is that in QG the ageostrophic flow can be diagnostically deduced from the balanced geostrophic flow, whereas in PE the ageostrophic flow is indirectly implied in the tendency equations of horizontal momentum and temperature. Consequently, the PE ageostrophic flow is much less clearly tied to the balanced geostrophic flow than in QG and contains, for example, baroclinic and barotropic inertia gravity waves (IGWs). We here want to stress that the difference between QG and PE ageostrophic flow is not primarily a matter of scale; for example, some of the IGWs have wave lengths and amplitudes that might seem consistent with the assumptions of QG, yet a QG model can still not represent them. Barotropic IGWs propagate very fast (> 300 m/s) and we thus need to adapt the time step in the PE model to accommodate them.

2. The values of the chosen parameters should be given (time step, resolution,...). Also all the values for the coefficients in the models should be given: r, D, alpha,.... In particular, some of the coefficients occur in the QG and the PE model (e.g. r and D). Are they the same in both models?

   The parameters are consistent between QG and PE as far as possible. In particular, model resolution and the physical coefficients are the same and only the time step differs to accommodate the fast propagation of IGWs in PE (cf. response to comment

1). We introduced a new table summarising all parameters and coefficients (new Tab. 3).

3. I think Bedymo could be very helpful to study the concept of balance. Recently, interest has increased significantly to better understand the coupling of the slow and the fast dynamics. QG solutions are balanced solutions without any internal gravity waves. Implementing such solutions into a PE model will destroy the balance. Bedymo would be an ideal tool for studying such processes, e.g. so called spontaneous imbalance. This means, however, to resolve the PE model properly. An overview on this issue can be found in a JAS special collection at https://journals.ametsoc.org/collection/spontaneous- imbalance .
We fully agree with this remark and thank the reviewer for bringing up this branch of the literature as context for our work. We include this perspective in the considerably extended introduction and the new outlook section (sec. 5).

4. The cases considered test mainly the quality of the QG model and the ability of the PE model to represent the QG solutions. Has the PE model also been tested against typical PE test cases?
We thank the reviewer for pointing out this slight gap in our evaluation. We introduced a new test case focussing on inertia-gravity waves excited by isolated orography in a mid-latitude channel in the revised manuscript (new sec. 3.1.4, Figs. 9+10). The results from Bedymo for this new test case generally agree well with the analytic solution.

5. The color code of Fig. 2, 4, 5 is not easy to read. Using different line styles might make the figures more clear.
We agree and adapted the line styles, colours and legends to make the Figures more intuitively readable and clear. In particular, we now consistently use colour for different categories and different line weights for gradual variations.

6. I think for the baroclinic life cycle and the storm track case the boundaries in the meridional direction are closed (v=0). However, for the Rossby wave case these boundaries seem to be open. Could the authors give a short description how the open radiative boundaries have been implemented in both models?
We unfortunately do not yet have the option for open boundaries in Bedymo. The model domain extends beyond the shown plot domain, as noted in the Figure caption.

7. What is the reason for the asymmetry in the PE Rossby wave case?
The slight asymmetries were introduced on one hand by IGWs excited by the shock of the initial flow adjusting to the topography, and on the other hand due to non-linearities in the flow. We reduced the mountain height by one order of magnitude in the revised version, such that non-linearities play a less important role. Further, we now mention the initial imbalance in the manuscript (L350-353).

8. For the coupled case a equatorial flow has been chosen that cannot be compared to QG dynamics that covers mid-latitude flows only. Instead the QG comparison, the PE

solutions are compared with analytical solutions from linear wave theory. This works only for sufficiently small heating. Was the local heating chosen comparable to the sources used e.g. in the paper by Gill (given in the reference)?

*Thanks for raising this point. The initial SST anomalies peaks at +5K, thus introducing a clearly finite-amplitude heating in the atmosphere. We now mention this caveat in the discussion of this test case (L397).*

For the linear theories, the ocean was passive. Was the motivation to couple an active surface layer ocean to study the differences?

*The reviewer is right, the variants of the test case with interactive ocean go beyond what could be captured in and thus evaluated against linear models. We nevertheless included them to showcase the additional dynamics introduced by several simple variants of a slab-ocean model. With the limitations of the linear model, we are here limited to plausibility checks—which we think the slab ocean model passes well. We now point out this caveat in the evaluation explicitly in the manuscript (L431-433).*

9. Recently, very interesting nonlinear solutions of equatorial waves have been documented (see e.g. Rostami, M., and Zeitlin, V. "Eastward-moving equatorial modons: a missing chain-link in the dynamics of the tropical atmosphere?", Phys. Fluids,31, 021701, 2019). For future work it would be very instructive to try to find such solutions with the PE model.

*We agree, and will keep this reference in mind for potential extensions of this work.*

**Comments from Anonymous Referee #2**

The study introduces a new modeling framework, aimed at solving the QG and PE equations as closely as possible. The QG model does not capture the dynamics outside the midlatitudes accurately and the primitive equation solvers are mostly formulated in the context of simulating the global circulation and are thus constructed differently making a direct comparison with a QG model elusive. The framework, designed in cartesian coordinates, is tested using 4 tests including both the atmosphere and the coupled atmosphere-ocean. The tests are carefully designed and succeed in demonstrating the strong similarities and differences between QG and PE by analyzing responses in baroclinic growth, coupling and stationary multi-wave forcings. I detail some fairly minor and some major comments below that can help improve the manuscript. I suggest accepting the manuscript, pending these revisions.

**Minor Comments**

1. Using sigma-coordinates in the vertical is common practice among models. Please either mention why is using sigma-cordinate in the vertical unique? Or remove this sentence altogether.

*We assume the reviewer refers to L4-5 in the abstract with this comment, where we write "As a consequence, but in contrast to most other quasi-geostrophic models, Bedymo is using sigma-coordinates in the vertical.". The use of sigma-coordinates is certainly common among PE models, but is very uncommon among QG models, which are most often formulated in pressure or log(pressure)-coordinates. The*

quoted statement is thus well justified.

2. L15 : "1950s"
3. L42 : as well → separately
4. L244 : should be "in line with"
5. In Figures 1,3,6,7 and 8, the x and y labels are interchanged. The x axis is the zonal distance and the y axis is the meridional distance. This should be corrected.
6. L246-7 : ... "results of Simmons and Hosking (1978), for instance, which" ...
Thanks for pointing out the above mistakes, we corrected them.

7. Section 3.1.2 : maybe I am missing something but how many levels are you using for the storm track test case? Were the original 26 vertical levels used for the PE case?
Thanks for pointing out this missing information. We added a table summarising model parameters to the manuscript (new Tab. 3). Unless noted otherwise (e.g. for the barotropic Rossby wave test case), the model was run with 3 vertical levels. This is in line with other typical QG setups (cf. for example Fromang and Riviere 2020, doi: 10.1175/JAS-D-19-0178.1).

8. I am confused by the writing here. There has to be zonal symmetries because you are computing the eddy covariances next. Please reconcile this. The stork tracks are not symmetric – so please revise the sentence.
Unfortunately, the line number to which this comment corresponds did not make it through in the review document. We assume you refer to L261 in the original submission, "Because of the absence of any zonal asymmetries, the resulting storm track is zonally symmetric." As far as we can see, this might be a misunderstanding as we here only talk about the "storm track", i.e. the eddy statistics aggregated over time, being zonally symmetric. This does not imply that, for example, the instantaneous wind velocity components are zonally symmetric as well, they are certainly not. We rephrased this sentence to "Because of the absence of any zonal asymmetries, the resulting storm track statistics are zonally symmetric." to avoid this misunderstanding (L293).

9. L276 : and the ones actually observed.
10. L296 : , however, do not
Thanks again for pointing out these mistakes, we corrected them.

**Major Comments**

1. I suggest a re-writing of the abstract because it is misleading with regards to the contents of the manuscript. There is no mention of the tracers in the manuscript, only the abstract. Thus, the sentence should be removed. There is no discussion about the graphical interface in the manuscript either, just a brief mention in the introduction section.
Thanks for pointing out this mismatch between abstract and manuscript. To resolve this, we removed the tracer module from the abstract and included a screenshot of the GUI in the manuscript (new Fig. 1).

2. The Introduction is inadequately written and I suggest revising it. No historical or existing literature has been discussed. It's severely lacking any conviction on the importance of the study and how it fits within recent modeling initiatives, apart from the mention that the models are being solved as closely as possible.

(a) Please discusses key differences between QG and PE and the traditional methods used to numerically solve them.

(b) Since this is the key foundation the manuscript stands on, please also elaborate the key differences between a QG model and the PE model, when used for atmospheric analysis (apart from the well-known fact that the QG framework works well only in the midlatitudes).

Thanks for pointing out these missing aspects in the introduction. We considerably extended the introduction, including a more comprehensive discussion of the different solution strategies in QG and PE, as well as two new paragraphs discussing physical and conceptual differences between these approximations.

(c) Horizontal and vertical coupling in models is challenging. Moreover, diffusion, which affects momentum balance strongly on long timescales, can also affect the model performance and is strongly dependent on the employed numerical schemes. In a QG model using sigma coordinates, it can introduce strong ageostrophic fluxes. Acknowledging these issues is important in the introduction and so is connecting to the other well established studies aimed at testing dynamical cores, to put the study into a proper perspective.

We agree with the reviewer in that numerical diffusion can be an issue that degrades model simulations. We have implemented in Bedymo advection schemes of 1st to 6th order, with the odd-order schemes being upwind biased. We have systematically tested the implementation of these schemes, for example using rotating a cone as a test case. In these tests, all implementations worked as expected, but of course lower-order schemes turned out considerably more diffusive than higher-order schemes. For our physical test cases, the choice of the advection scheme however turned out largely inconsequential as the explicit diffusion is dominating over the numerical diffusion. We thank the reviewer for pointing out this gap in the description of the model numerics. We added a brief discussion on this issue (L184-186).

I recommend adding some discussion connecting this study to the more recent studies on evaluation of dynamical cores in much more detail :

- Lin, S.-J., Harris, L., Chen, X., Yao, W. & Chai, J. (2017). Colliding modons: A nonlinear test for the evaluation of global dynamical cores. Journal of Advances in Modeling Earth Systems, 9, 2483 2492. https://doi.org/10.1002/2017MS000965
- Gupta, A, Gerber, EP, Lauritzen, PH. Numerical impacts on tracer transport: A pro- posed intercomparison test of Atmospheric General Circulation Models. Q J R Meteorol Soc. 2020; 146: 3937 3964. https://doi.org/10.1002/qj.3881
- Ma, J., Xu, S., & Wang, B. (2020). Reducing numerical diffusion in dynamical coupling between atmosphere and ocean in Community Earth System Model version 1.2.1. Journal of Advances in Modeling Earth Systems, 12, e2020MS002052. https://doi.org/10.1029/2020MS002052

- Held, I. M., & Suarez, M. J. (1994). A Proposal for the Intercomparison of the Dynamical Cores of Atmospheric General Circulation Models, Bulletin of the American Meteorological Society, 75(10), 1825-1830.

We thank the reviewer for these pointers. We considerably extended the discussion of Held and Suarez (1994) (L309-314, cf. also response to point 3 below). We did however not include the other references, as we decided against taking up the colliding modons as another new test case. Further, as argued above, numerical diffusion is not a primary concern for Bedymo, and, as the reviewer noted, we leave Bedymo's tracer module unevaluated in the present manuscript. We thus now only mention the tracer module it the new outlook section (sec. 5).

3. Since the model is developed ground up and prepared for educational and research purposes, it is important to provide a performance analysis with a traditional dynamical core. Thus, I suggest the authors to add another a discussion/figure and/or qualitatively compare the runtime of Bedymo with a Held-Suarez like model, for instance, with similar numerical schemes.

We would have very much liked to do such a comparison, but contrary to intuition there are no other established dynamical cores that are similar enough to make such a comparison insightful. Held-Suarez-type models are global, and thus include a Hadley circulation and thus subtropical jets, whereas the jets in Bedymo by construction can only be eddy-driven. Further, Held-Suarez-type models need to take into account the spherical geometry of the Earth, which Bedymo does not. For these reasons, a comparison of, for example the wind speed in Figure 2 of Held and Suarez (1994) with our Figure 4 cannot really evaluate whether Bedymo is doing the right thing for the right reason. Nevertheless, we agree with the reviewer that the current discussion of Held-Suarez is too superficial. Hence, we expanded the discussion and now take up the above arguments in the manuscript (L309-314).

4. The study would make for a much more valuable contribution to the existing literature if there was a functionality to use the PE framework in spherical coordinates as well, considering the overall curvature of the planet. Is that a possibility with the framework so far? If not, then the use of the model might be quite limited.

For now, Bedymo does not support a spherical geometry. An extension towards spherical geometry is one of our plans for further development and is mentioned it in the new outlook section (sec. 5).

5. I strongly encourage the authors to elaborate more on the future possibilities using the model in the conclusion/discussion section. Is there a plan to introduce more features in the model (this might be a good point to talk briefly about the tracer module)? How does having a "live" graphical interface in the model give it an edge over the traditional frameworks (even modern dynamical cores from prominent modeling centered allow visualisation using certain software packages)?

We agree with the reviewer, and included these points
   a. by including and discussing a screenshot of the graphical user interface in the manuscript (new Fig. 1),

b.  by a new outlook section detailing plans and perspectives for future development (sec. 5), and
c.  by providing and citing a separate user guide that describes how to setup and run the model (DOI: 10.5281/zenodo.5909781).